



# Impact of reduced emissions on direct and indirect aerosol radiative forcing during COVID–19 lockdown in Europe

Simon F. Reifenberg[1,a], Anna Martin[1], Matthias Kohl[1], Zaneta Hamryszczak[1], Ivan Tadic[1],
Lenard Röder[1], Daniel J. Crowley[1], Horst Fischer[1], Katharina Kaiser[2], Johannes Schneider[2],
Raphael Dörich[1], John N. Crowley[1], Laura Tomsche[3], Andreas Marsing[3], Christiane Voigt[3,4],
Andreas Zahn[5], Christopher Pöhlker[6], Bruna A. Holanda[6], Ovid Krüger[6], Ulrich Pöschl[6],
Mira Pöhlker[6,7,8], Patrick Jöckel[3], Marcel Dorf[1], Ulrich Schumann[3], Jonathan Williams[1],
Joachim Curtius[9], Hardwig Harder[1], Hans Schlager[3], Jos Lelieveld[1,10], and Andrea Pozzer[1]

[1]Max Planck Institute for Chemistry, Atmospheric Chemistry Department, 55128 Mainz, Germany
[2]Max Planck Institute for Chemistry, Particle Chemistry Department, 55128 Mainz, Germany
[3]Deutsches Zentrum für Luft– und Raumfahrt, Institute for Atmospheric Physics, Oberpfaffenhofen, 82230 Wessling,
Germany
[4]Johannes Gutenberg–University Mainz, Institute for Physics of the Atmosphere, 55099 Mainz, Germany
[5]Institute of Meteorology and Climate Research, Karlsruhe Institute of Technology, 76344 Karlsruhe, Germany
[6]Max Planck Institute for Chemistry, Multiphase Chemistry Department, 55128 Mainz, Germany
[7]Faculty of Physics and Earth Sciences, Leipzig Institute for Meteorology, University of Leipzig, 04318 Leipzig, Germany
[8]Experimental Aerosol and Cloud Microphysics Department, Leibniz Institute for Tropospheric Research,04318 Leipzig,
Germany
[9]Goethe–University of Frankfurt, Institute for Atmospheric and Environmental Sciences, 60438 Frankfurt am Main, Germany
[10]The Cyprus Institute, Climate and Atmosphere Research Center, Nicosia, 1645, Cyprus
[a]now at: MARUM – Center for Marine Environmental Science, University of Bremen, Bremen, Germany

**Correspondence:** Andrea Pozzer (andrea.pozzer@mpic.de)

**Abstract.**

Aerosols influence the Earth's energy balance through direct radiative effects and indirectly by altering the cloud microphysics. Anthropogenic aerosol emissions dropped considerably when the global COVID–19 pandemic resulted in severe restraints on mobility, production, and public life in spring 2020. Here we assess the effects of these reduced emissions on direct

and indirect aerosol radiative forcing over Europe, excluding contributions from contrails. We simulate the atmospheric composition with the ECHAM5/MESSy Atmospheric Chemistry (EMAC) model in a baseline (business as usual) and a reduced emission scenario. The model results are compared to aircraft observations from the BLUESKY aircraft campaign performed in May/June 2020 over Europe. The model agrees well with most of the observations, except for sulfur dioxide, particulate sulfate and nitrate in the upper troposphere, likely due to a somewhat biased representation of stratospheric aerosol chemistry

and missing information about volcanic eruptions which could have influenced the campaign. The comparison with a business as usual scenario shows that the largest relative differences for tracers and aerosols are found in the upper troposphere, around the aircraft cruise altitude, due to the reduced aircraft emissions, while the largest absolute changes are present at the surface. We also find an increase in shortwave radiation of $0.327 \pm 0.105 \ \mathrm{Wm^{-2}}$ at the surface in Europe for May 2020, solely attributable to the direct aerosol effect, which is dominated by decreased aerosol scattering of sunlight, followed by reduced



aerosol absorption, caused by lower concentrations of inorganic and black carbon aerosols in the troposphere. A further increase in shortwave radiation from aerosol indirect effects was found to be much smaller than its variability. Impacts on ice crystal- and cloud droplet number concentrations and effective crystal radii are found to be negligible.

## 1 Introduction

Aerosols play a pivotal role in both air pollution and climate change. They cause millions of years of lost life expectancy per
year globally (Lelieveld et al., 2015, 2020), impose a negative (net) effective radiative forcing (Bellouin et al., 2020), and are a large source of uncertainty in climate change assessments. A reduction of the cooling effect by a decreased aerosol burden necessitates stronger reductions of greenhouse gases (GHGs) for a targeted net radiative forcing (Larson and Portmann, 2019).

Owing to the central importance of aerosol particles, the reduced emissions resulting from drastic restrictions on mobility, industry and public life during the COVID–19 "lockdowns" in early 2020 (here referred to as "lockdown") (Barré et al., 2020;
Evangeliou et al., 2021; Guevara et al., 2021; Le Quéré et al., 2020) sparked a plethora of publications on the subsequent effects on local, regional, and global air pollution (see, for instance, He et al., 2020; Liu et al., 2020; Petetin et al., 2020; Tobías et al., 2020; Venter et al., 2020; Mertens et al., 2021) and climatic effects, on which we will focus in this work.

We recognise that reduced emissions during lockdown do not necessarily translate into improved air quality, as primary pollutants take part in a complex set of chemical processes, which need to be included in a thorough analysis (Kroll et al., 2020).
For instance, although ozone was reported to be reduced in the free troposphere in the northern hemisphere (Steinbrecht et al., 2021), the reduced emissions of the nitrogen oxides NO and $NO_2$ led to an increase in ozone concentrations in urban locations, as an important short-term sink (reaction with NO) was reduced (e.g. Gkatzelis et al., 2021; Sicard et al., 2020; Mertens et al., 2021). This illustrates how the complex (photo-)chemistry and the nonlinearity of the underlying chemical system have to be described and analyzed within the framework of a dynamic atmospheric chemistry model. A chemistry climate model with
appropriate chemistry furthermore enables a direct comparison of business as usual and reduced emissions within the same synoptic background conditions, complementary to a purely observation-based approach. Such model investigations are most effective when accompanied and guided by observational data, as will be done in the present study.

The interaction of aerosols with radiation and their climatic impact can be categorized into two types: (i) direct effects by impact on radiation fluxes, and (ii) indirect effects through changes in cloud physical and optical properties. The direct effects
include absorption and scattering of electromagnetic waves, whereby aerosol particles, most prominently black carbon (BC), absorb incoming solar radiation, which leads to warming of the ambient air and decreases solar irradiance in the layers below. In addition, aerosols scatter incident radiation back to space, leading to a net cooling of the climate system on average. These processes depend on the size, shape and chemical composition of the aerosols and on the wavelength of the radiation. In addition, the net effect depends on the surface albedo (Yoon et al., 2019; Bellouin et al., 2020).
The reduced emissions in spring 2020 are thus expected to affect aerosol radiative forcing. A reduction in the backscattering of solar radiation is expected to result in warming, which is offset by the anticipated cooling effect through a reduction of black carbon emissions, and the net effect may vary vertically and horizontally. For instance, Gettelman et al. (2021) reported





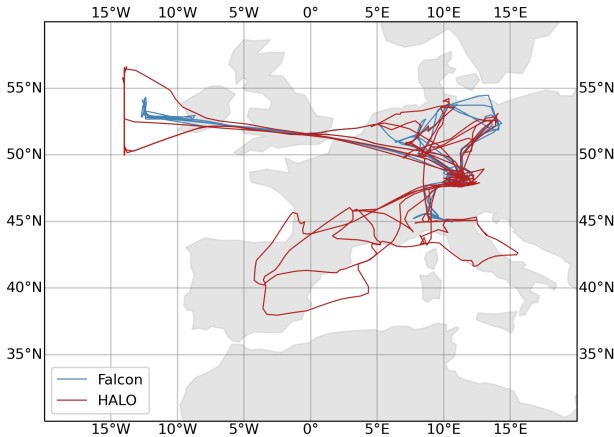

**Figure 1.** Tracks of conducted flights during the BLUESKY campaign (16th May to 9th June 2020). Colors denote the aircraft, Falcon (blue) and HALO (red).

a simulated net warming at the surface and in the lower troposphere in most regions, caused by enhanced insolation at the surface, and cooling in upper layers of the troposphere, due to reduced absorption by black carbon. They also determined a

difference in the clear sky net shortwave (SW) flux at the top of the atmosphere (TOA) of up to $0.1 \ \mathrm{Wm^{-2}}$ globally in May between simulations with and without reduced emissions, i.e. less outgoing SW radiation due to the lockdown. Complementing the analyses regarding these more immediate effects, Forster et al. (2020) estimate a short-term warming driven by a weakened aerosol cooling through reduced sulfur dioxide ($SO_2$) emissions, followed by a cooling of $0.010{\pm}0.005$ K by 2030 in reference to a baseline scenario.

In addition to the aerosol direct effects on the radiation budget, aerosol particles can trigger several indirect effects. Aerosol particles serve as cloud condensation nuclei and thus alter cloud properties, such as cloud albedo, cloud droplet number concentration, formation processes, precipitation and cloud lifetime (see, for instance, Bellouin et al., 2020; Christensen et al., 2020; Lohmann and Feichter, 2005; Twomey, 1959). In turn, clouds also affect aerosols. Clouds convert precursor gases into aerosol particles through heterogeneous chemistry (Ervens et al., 2011; Lelieveld and Heintzenberg, 1992; McMurry and Wil-

son, 1983) and, at the same time, remove aerosols and soluble gases from the atmosphere by precipitation ("wet deposition").

Clouds are classified based on their liquid water or ice content, since these characteristics determine the efficiency of reflection of solar radiation back to space (cloud albedo effect) and the absorption rate of longwave radiation (greenhouse effect), thus their impact on the Earth's radiation budget. Clouds consisting only of ice crystals (e.g. cirrus clouds) occur mostly in high altitudes at temperatures below the homogeneous freezing threshold of $\sim -35°\mathrm{C}$ (Pruppacher and Klett, 1997; Kärcher

and Seifert, 2016; Bacer et al., 2021). Ice crystals (ICs) are formed either due to homogeneous or heterogeneous freezing depending on aerosol characteristics and number concentration, temperature, supersaturation and vertical air motion (Pruppacher and Klett, 1997). IC number densities in cirrus clouds are typically lower (Voigt et al., 2017) compared to clouds at lower altitude, and cirrus clouds are often optically thin and transmit shortwave radiation and absorb longwave radiation, resulting in





an overall positive net radiative effect at the TOA (Bacer et al., 2018, 2021; Gasparini et al., 2017). In contrast, mixed-phase
clouds, consisting of ICs and cloud droplets, form only due to heterogeneous nucleation at lower altitudes and temperatures
higher than $-35°C$. These optically thick clouds reflect a comparably large amount of the incoming SW radiation, leading to
a negative net radiative effect at the TOA (Chen et al., 2000). The cloud albedo effect can be enhanced by the Twomey effect
(Twomey, 1959), whereby high number concentrations of CCN lead to a higher number of droplets of smaller size compared
to clouds formed with lower number of CCN. Accordingly, clouds, containing more droplets of smaller sizes, reflect solar
radiation more strongly than clouds containing fewer droplets of larger size. This effect is especially important over the ocean,
where the number of natural CCNs, compared to anthropogenic CCNs, is much lower than over land (Lohmann and Feichter,
2005). Ship emissions, including aerosol particles, create clouds with a much higher reflectivity (or albedo) than clouds formed
by the less numerous natural CCNs outside of the ship tracks (Platnick et al., 2000). The same effect arises in aircraft flight
tracks, where the emitted particles may act as ice nucleating particles, leading to contrail formation (Schumann et al., 2017)
and an increase in cirrus cloud formation in the upper troposphere (Boucher, 1999). Still, the magnitude of the aviation soot
effect on the radiation budget remains uncertain (Urbanek et al., 2018; Righi et al., 2021). Recently, satellite data have been
used to quantify changes in clouds in regions with COVID-reduced airtraffic in 2020 (Quaas et al., 2021; Gettelman et al.,
2021). With respect to contrails, Schumann et al. (2021a, b) find a substantial reduction of contrail cirrus optical thickness and
radiative forcing during the lockdown period.

The modification of cloud cover resulting from differing particle number concentrations has further effects on the lifetime
of a cloud. An increase in smaller cloud particles makes the clouds more persistent and delays the occurrence of precipitation
(referred to as cloud lifetime effect, Lohmann and Feichter, 2005). Increased cloud lifetime affects not only the timing but also
the location of precipitation. The increased lifetime of clouds also extends the period over which a cloud can reflect radiation,
thus adding to the Twomey effect. With increasing aerosol number concentration corresponding to a negative perturbation
of the radiation budget, both effects contribute to cooling at the surface and at the TOA (Lohmann and Feichter, 2005). The
number and size distribution of cloud nucleating particles can thus affect the extent of transmission of radiation.

Using an observation-guided model, the COVID–19 lockdown provided an opportunity to examine how the climate system
reacts to perturbations such as abruptly reduced air pollution emissions. The COVID–19 lockdown may also serve to assess the
impact of economic recovery with respect to climate change mitigation: for instance, Forster et al. (2020) show that investments
aimed at a "green" opposed to a fossil-fueled recovery can reduce projected warming by $0.3$ K by 2050, with only negligible
contributions from the lockdown.

In the present study, we simulate the chemical composition of the atmosphere in Europe in spring 2020 under a reduced
emission scenario and a business as usual scenario with a state-of-the-art climate and chemistry simulation system, constraining
atmospheric dynamics by reanalysis meteorological data. We use a unique observational data set of trace gases and aerosols
obtained during an aircraft measurement campaign in Europe during the COVID–19 lockdown in summer 2020 to evaluate the
model results. We then quantify the effects of the lockdown on radiative transfer in the atmosphere, particularly the change in
shortwave fluxes and shortwave heating rates attributable to a reduced aerosol burden in Europe. Furthermore, we examine the





impacts of the lockdown scenario on cloud properties, including potential changes of the radiative forcing caused by indirect aerosol effects.

This paper is structured as follows. In Sect. 2 we describe the model together with an overview of the simulations performed (Sect. 2.1), as well as the observational data (Sect. 2.2). The model evaluation is presented in Sect. 3. We then investigate the impacts of the reduced emissions during the 2020 COVID–19 lockdown on direct aerosol effects and indirect cloud–aerosol effects (Sect. 4).

## 2   Data and methods

### 110   2.1   Model data

The ECHAM5/MESSy Atmospheric Chemistry (EMAC) model is a numerical chemistry and climate simulation system that includes submodels describing tropospheric and middle atmospheric processes and their interaction with oceans, land and human influences (Jöckel et al., 2016). It uses the second version of the Modular Earth Submodel System (MESSy2) to link multi-institutional computer codes. The core atmospheric model is the 5th generation European Centre Hamburg general

circulation model (ECHAM5, Roeckner et al., 2006).

    For the present study we applied EMAC (ECHAM5 version 5.3.02, MESSy version 2.55.0) in T63L47MA-resolution, i.e. with a spherical truncation of T63 (corresponding to a quadratic Gaussian grid of approx. 1.8 by 1.8 degrees in latitude and longitude) with 47 vertical hybrid pressure levels up to 1 Pa. Roughly 22 levels are included in the troposphere. The dynamics of the EMAC model has been weakly nudged (Jeuken et al., 1996; Jöckel et al., 2006) towards the ERA–interim meteorological

reanalysis data (Berrisford et al., 2011) of the European Centre for Medium-Range Weather Forecasts (ECMWF) to represent the actual day to day meteorology in the troposphere.

    The setup of the chemistry submodels for this study is similar to the one presented by Jöckel et al. (2016, simulation RC1–aero–07), but with the addition of the submodel ORACLE (Tsimpidi et al., 2014) for the organic chemistry calculation and with stratospheric heterogeneous chemistry neglected. Initial conditions for the meteorology were also taken from the ERA–interim

reanalysis data, while the ones for the chemical composition were from previous EMAC simulations (Pozzer et al., 2021). In addition, the anthropogenic emissions used are based on CAMS–GLOB–ANTv4.2 (Granier et al., 2019). To reproduce the effect of lockdown on the emissions, we adopted the reduction coefficient for Europe as in Guevara et al. (2021) for the sectors of energy production (ENE), road transport (TRO) and industrial processes (IND). The reduced emissions were averaged for the period 19th April to 26th of April (i.e. last available week in the dataset), and applied (for each country) for March, April,

May and June. For aviation (AVI) we adopted the same method, although we applied the estimated factor to the entire aviation emissions, without any country distinction.

    The aerosol–cloud interactions are based on the aerosol microphysics parameterization of Pringle et al. (2010) including aerosol aging and the continuous calculation of aerosol number concentration depending on the mass mixing ratio and mixing state. Additionally, cloud formation processes are accounted for using the convection framework of Tost et al. (2006) based

on the convection calculation schemes of Tiedtke (1989) and Nordeng (1994). Large-scale cloud formations and prognostic





variables depending on cloud microphysical processes follow the work of Lohmann et al. (2007); Lohmann and Hoose (2009); Bacer et al. (2018). Thereby, the ice crystal number concentration (ICNC) formation in the cirrus regime ($T \leq 238.15$ K) was modified according to Neubauer et al. (2019) to reduce the artificial homogeneous freezing of dry aerosol particles independent of availability of water vapor.

We performed four simulations, all covering the period from January 2019 to July 2020:

– STD : standard (i.e. "business as usual") emissions, without cloud–aerosol interaction,

– RED : reduced emissions due to lockdown, without cloud–aerosol interaction,

– STDCLOUD : like STD but with aerosol–cloud interaction,

– REDCLOUD : like RED but with aerosol–cloud interaction.

In all simulations performed, the impact of different aerosol concentrations on the radiation (discussed in Sect. 4.2.1) is diagnosed but not used by the general circulation model, which instead adopts an aerosol climatology (Pringle et al., 2010). Similarly, changes in the tracers (e.g. ozone) do not influence the radiation, which is calculated with a greenhouse gases climatology.

The model evaluation is performed with the RED simulation, while its difference with the STD simulation is used to evaluate the impact of the reduced emissions during the lockdown. Simulation RED and STD have binary identical dynamics (Deckert et al., 2011), as no feedback between chemistry and dynamic is present. Differently, in REDCLOUD and STDCLOUD, the aerosol–cloud interaction is activated following the work of Lohmann and Hoose (2009); Bacer et al. (2018), leading to modification of cloud properties and therefore to changes in radiation and dynamics. The simulations REDCLOUD and STDCLOUD are only used for estimating the indirect effects of aerosols (see Sect. 4.2.2).

## 2.2 BLUESKY observational data

We compare simulated trace gas and aerosol abundances to a comprehensive set of observations obtained during the BLUESKY campaign (Voigt et al., 2021), led by the German Aerospace Center (DLR) and the Max Planck Institute for Chemistry (MPIC), with the aim of investigating the effects of reduced emissions on atmospheric chemistry and physics. From 16th May to 9th June 2020 in situ measurements of trace gases and trace particles were conducted in the atmosphere over European urban areas and the North Atlantic flight corridor with the High Altitude and Long Range (HALO) research aircraft and a second research aircraft, Falcon (see Fig. 1 for flight paths). In total 8 and 12 flights were conducted with the HALO and the Falcon, respectively.

We compare aerosol mass concentrations of black carbon (BC, size range between 70 and 500 nm), sulfate ($SO_4^{2-}$), nitrate ($NO_3^-$), ammonium ($NH_4^+$), organic aerosol particles (ORG, all from 40 to 800 nm) and aerosol particle number concentrations (between 250 nm to 40 μm). These are complemented by volume mixing ratios of carbon monoxide (CO), ozone ($O_3$), nitric oxide (NO), hydrogen peroxide ($H_2O_2$), peroxyacetyl nitrate (PAN), nitric acid ($HNO_3$), and sulfur dioxide ($SO_2$). Details regarding instrumentation are provided by Voigt et al. (2021). We additionally use air temperature $T$, wind speed and specific





humidity $q$ to assess the quality of the reproduced synoptic conditions which are constrained (nudged) in the model. For the comparison, the model output was sampled online in space and time by the submodel S4D (Jöckel et al., 2016), following the
flight tracks of the field campaign and with a time frequency of 5 minutes.

## 3   Results: Model evaluation

A summary of the comparison of observations and model results is listed in Table 1 and presented graphically in Figs. 2 and 3.
The ambient air temperature $T$ is reproduced very well by the model; the average ratio of observed and simulated $T$ is equal to 1.00 with a normalized root mean squared error of 0.04 (NRMSE; RMSE divided by range of observations). The vertical
temperature profile is matched in the lower and free troposphere with a slight underestimation of observed temperatures towards the upper troposphere (Fig. 2). Specific humidity $q$ is also captured reasonably well in the model, (NRMSE = 0.06), as 85.9 % of simulated values lie within a factor of two of the observations, yet slightly overestimated (see Fig. 2 and Table 1). In addition, horizontal wind speed $||\boldsymbol{u}_h||$ is also reproduced accurately with a low NRMSE (0.06) and an average ratio of 1.02.

Overall, the agreement between the meteorological variables from model and observations indicates successful initialization
and nudging of meteorological variables and that the meteorological conditions during the relevant time period are simulated adequately. As the model is not nudged in the stratosphere or boundary layer (the nudging coefficient is maximal in the free troposphere (Jöckel et al., 2006)), the slight underestimation of temperature in the upper troposphere region is not surprising. Nevertheless, the temperature bias is much lower than in other EMAC studies, despite the use of same nudging method and coefficients (Jöckel et al., 2016), due to the intialization and shorter simulation time in this work. As temperature and humidity
are important quantities regarding cloud formation, and accurate wind vectors are key for representing advective processes, the following analyses of atmospheric composition and the effects on radiative transfer build on an accurate representation of the meteorological state of the model.

### 3.1   Trace gases

Observed ozone ($O_3$) mixing ratios are reproduced well by the model. More than 94 % of simulated values are within a factor
of 2 of the observations ("PF2" value), the normalized root mean squared error of 0.04 is low. Nevertheless, the model seems to slightly overestimate the observations, as already pointed out in various studies (e.g. Jöckel et al., 2016).

Simulated carbon monoxide (CO) mixing ratios are also in a good agreement with the observations, and virtually all simulated values lie within a factor of two of the observations. However, especially at lower altitudes, the simulated mixing ratios somewhat underestimate the observed values, although the difference between average observations and average model results
are well within their respective variability, and the shape of the vertical profile is qualitatively well reproduced. The same holds for nitric oxide (NO), which exhibits a C-shaped profile. The NRMSE for NO is low (0.08) and the average ratio of simulated to observed mixing ratio is 0.99, however more than a third of simulated values deviate more than a factor of two from the observations, due to the high variability of this tracer. Particularly the range of the observed mixing ratios close to the surface





**Figure 2.** Vertical distribution of simulated (red, "MOD") and observed (blue, "OBS") tracer mixing ratios and two meteorological variables ($T$ and $q$), represented by box–whisker plots for pressure bins. The white line marks the median, the box corresponds to lower and upper quartiles, the whiskers represent the 5–95 percentile. The grey numbers on the right indicate the sample size (number of observed and interpolated simulated data points) for each pressure bin. Simulated values are from the RED simulation, i.e. with reduced emissions and no aerosol–cloud interactions. For $HNO_3$ and $SO_2$ (measured onboard of the Falcon aircraft, grey number marked with asterisks) the domain average of the model results over Europe at the corresponding altitude were used, not the values sampled online on the flight track.



is not well reproduced by the model, which results from the short lifetime of NO and the challenge in reproducing its local variation by a global model.

Hydrogen peroxide ($H_2O_2$) and peroxyacetyl nitrate (PAN) are less well represented, the average ratios of simulated to observed mixing ratio (2.01 for $H_2O_2$ and 1.91 for PAN) indicate an overestimation by the model. Nevertheless, for both species about two thirds of the simulated points are still within a factor of two of the observations (see Fig. 2), and the measured dependence on altitude is captured by the model.

Sulfur dioxide ($SO_2$) was sampled predominantly at high altitudes between 370 to 170 hPa, where it is strongly underestimated by the model. We hypothesize that the systematic underestimation of $SO_2$ concentrations is due to an inaccurate representation of transport from the boundary layer or from the stratosphere to the upper troposphere or due to model shortcomings within the stratospheric aerosol chemistry, which will be discussed briefly as part of the following Sect. 3.2. All in all, as summarized in Table 1, there is reasonable agreement between observed and simulated mixing ratios of the trace gases investigated.

## 3.2 Aerosols

The vertical profile of the aerosol number concentration is qualitatively reproduced (see Fig. 3). In the lowest altitude pressure bin, the range and median of the observations and model results match very well. There are some deviations between 850 and 480 hPa, where simulated number concentrations are larger than the observed ones, although this overestimation is well within the observations' variability. This overestimation dominates the average ratio of modeled to measured values (2.60, see Table 1).

The measured black carbon (BC) concentrations are captured well by the model close to the surface, while the observational variability is underestimated at high altitudes. The NRMSE of 0.09 is relatively low, as the higher abundance closer to the surface – that is, closer to the sources – is well represented, both in terms of magnitude and variability.

Sulfate ($SO_4^{2-}$) exhibits qualitatively similar features as BC; the relatively high concentrations observed in the lower troposphere are matched by the simulated concentrations, yet there is a significant underestimation of sulfate aerosol concentrations in the upper troposphere. We hypothesize that the modeled underestimation of sulfur species is related to missing contribution of volcanic eruptions that have reached the stratosphere at low latitudes and return to the troposphere at higher latitude. Many small and medium size eruptions have been reported in the year prior to the BLUESKY campaign (https://volcano.si.edu, last access 30. October 2021), but their influence on the upper troposphere and lower stratosphere is yet to be quantified. Some preliminary test simulations are mentioned below.

Between 1050 and 625 hPa simulated organic aerosol concentrations are somewhat larger in the model than in reality, the shape of the vertical profile is, however, qualitatively reproduced.

Nitrate ($NO_3^-$) and ammonium ($NH_4^+$) concentrations close to the surface are generally well reproduced. While, at higher altitudes, the simulated $NH_4^+$ agrees with the observations, simulated nitrate is too high, which is probably related to the co-located underestimation of sulfate.





**Table 1.** Summary of model–observations comparison. The same spatio-temporal location were used for all simultaneously available points . NRMSE shows the root mean squared error normalized by the range of the observations. PF2 denotes the percentage of model points within a factor of 2 of the observations. The column $\overline{\text{MOD/OBS}}$ is the average of the simulated and observed data ratios.

| Variable | NRMSE | PF2 | $\overline{\text{MOD/OBS}}$ |
|---|---|---|---|
| *Trace gases* | | | |
| $O_3$ | 0.04 | 94.7 | 1.25 |
| CO | 0.14 | 99.3 | 0.98 |
| NO | 0.08 | 65.0 | 0.99 |
| $H_2O_2$ | 0.32 | 61.5 | 2.01 |
| PAN | 0.13 | 60.3 | 1.91 |
| $HNO_3$ | 0.37 | 12.9 | 0.46 |
| $SO_2$ | 0.40 | 25.9 | 0.43 |
| *Aerosols* | | | |
| BC | 0.09 | 18.6 | 0.68 |
| $NO_3^-$ | 0.14 | 20.6 | 0.92 |
| $NH_4^+$ | 0.22 | 28.8 | 0.83 |
| $SO_4^{2-}$ | 0.16 | 26.8 | 0.72 |
| Organics | 0.45 | 40.6 | 1.73 |
| Number conc. | 0.11 | 42.8 | 2.60 |
| *Meteorology* | | | |
| $T$ | 0.04 | 100.0 | 1.00 |
| $q$ | 0.06 | 85.9 | 1.27 |
| $\|\boldsymbol{u}_h\|$ | 0.06 | 100.0 | 1.02 |

The results of the model/measurement comparison are summarised in Table 1. We conclude that there is generally reasonable agreement between simulated and observed trace gases and aerosols with some deviation of the aerosol concentrations, especially in the mid-upper troposphere.

A single factor causing the model underestimation of BC and sulfate aerosol concentrations in the upper troposphere, e.g. a localized plume of pollution, is judged unlikely, as BC and $SO_4^{2-}$ do not correlate ($r < 0.01$, $p = 0.90$): in fact, mapping observed $SO_4^{2-}$ concentrations to ozone (a tracer of stratospheric air), and carbon monoxide (a tracer of tropospheric air), reveals that high $SO_4^{2-}$ concentrations coincide with high ozone ($r = 0.83$, $p < 0.01$) and low carbon monoxide ($r = -0.65$, $p < 0.01$) (Fig. 4). A similar, yet weaker, correspondence can be found in the simulated data (see Fig. 4). This suggests a stratospheric

source of sulfate aerosols in both model and reality, although stratospheric sulfate aerosol appears to be represented poorly in the model. It is noteworthy that a precursor for sulfate aerosols, sulfur dioxide, is also systematically underestimated. We





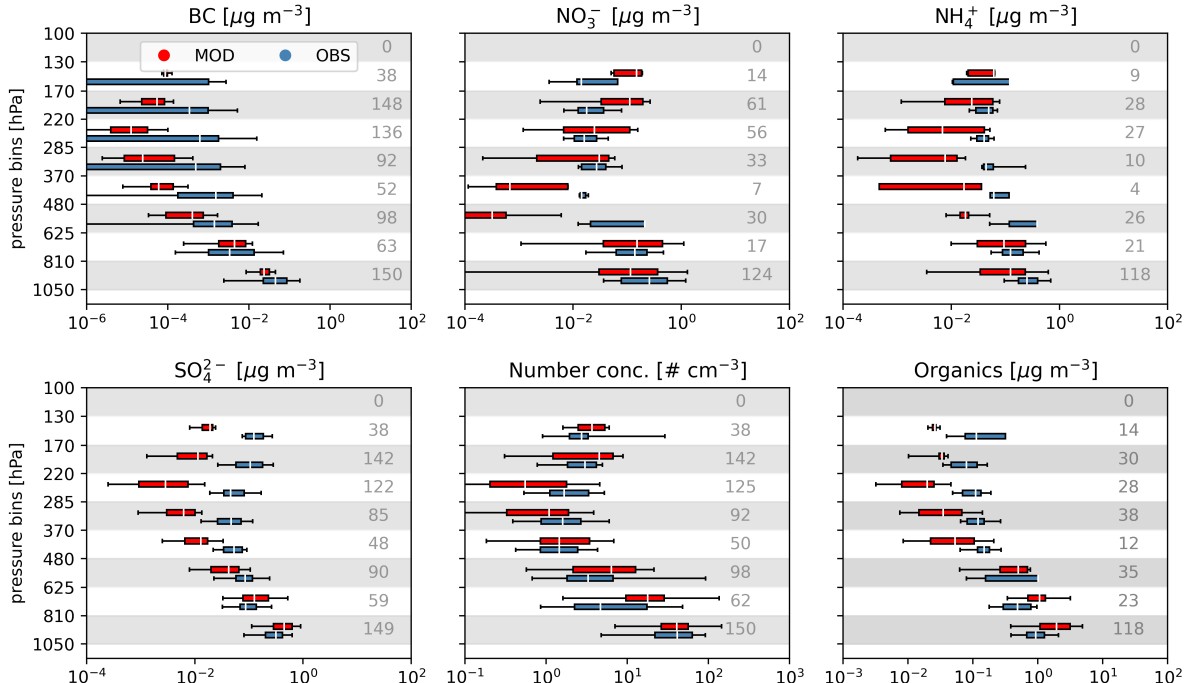

**Figure 3.** As Fig. 2, but for aerosols. Please note the logarithmic scale in the x-axis.

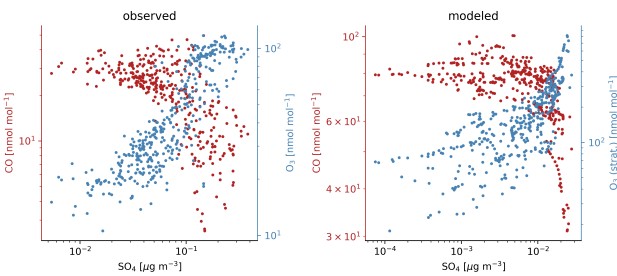

**Figure 4.** Scatter plot of co-located $SO_4$ and CO (red) respective $O_3$ (blue) abundance between 350 and 150 hPa from observations (left) and the RED simulation (right).

assume that the high $SO_2$ abundance measured is of volcanic origin. We tested this by injecting high levels of $SO_2$ in the stratosphere in additional simulations, mimicking volcanic eruptions at Raikoke (June) and Ulawun (June and August) in 2019 (see de Leeuw et al., 2021; Kloss et al., 2021). However, this did not affect the concentrations of $SO_4^{2-}$ and $NO_3^-$ significantly (not

shown). A partial increase of $SO_4^{2-}$ was obtained by including the volcanic eruption of Taal in January 2020. Nevertheless this is still not enough to bring the model results close to the observations. We therefore conclude that our observed underestimation of $SO_4^{2-}$ is of stratospheric origin, although it is not fully clear what caused it.





A further partition of the region of interest into three subregions (Central Europe, Southern Europe, Atlantic) did not reveal substantial spatial dependencies of model deviation from observations (not shown).

## 4 Results: Impact of reduced emissions

To quantify the effect of the lockdown, we use the business as usual simulations (STD and STDCLOUD) in the analyses. We focus on May 2020, as this time period is covered by the measurement campaign and the atmosphere can be expected to have adjusted to the impact of abruptly reduced emissions. We also analyse the impact in an area encompassing Europe (the region of study), i.e. over a longitude-latitude box from $-20$ to $20°$ E and $30$ to $60°$ N (exactly the depicted map sector in Fig. 1).

### 4.1 Impact on tracers and aerosols

As no feedback between chemistry and dynamics is activated in the RED and STD simulations, any differences between these simulations are purely attributable to the different emissions during the lockdown and the consequent different chemical regimes.

In general, while large absolute changes are expected at the surface, in the upper troposphere (UT) we find the largest relative changes, due to the strong influence of the local emissions and to the low mixing ratios of most of the species investigated (see Fig. 5). Large relative changes in the UT are found for NO, $SO_2$ and BC, with a strong reduction ($\sim 50$ % or more) in the region between $200$ and $300$ hPa, i.e. the typical aircraft cruise altitude. The reduced air traffic during the lockdown period greatly decreased the emissions of nitrogen oxides into the UT, and the effects of the lockdown on other tracers in the UT are mostly a result of this strong reduction. Hydroxyl radicals (OH) decrease by roughly $20$ % in the UT and $5$ % elsewhere in the troposphere, a direct effect of a reduced OH recycling by $NO_x$. Despite the reduced OH, carbon monoxide does not increase, due to the decrease in the direct emissions. The overall effect of the lockdown for most tracer is a combination of reduced emissions and reduced sinks (i.e. oxidation via OH): while this is well balanced for CO (changes in the order of few percent), for $SO_2$ the emission reductions are larger than the decrease in the reaction with OH, causing its mixing ratio to be reduced (up to $50$ % in the UT) compared to the business as usual scenario. It must be stressed however, that those relative changes in the upper troposphere, although significant, have a very minor impact on most trace gas budgets, due to their low mixing ratios at these altitudes.

Similar to the trace gases, for most aerosols the lockdown reduces their concentration mostly at the surface, although the largest relative differences are simulated in the UT, due to the low concentration at these altitudes. For example, sulfate is subject to a large relative change in the UT but to much larger absolute changes close to the surface, mimicking the changes in $SO_2$ (see also Fig. 6). Furthermore, BC decreases significantly in the whole troposphere, due to the strong reduction of the emissions both at the surface and in the UT (from aircraft). The aerosol reductions during the lockdown have implications on the incoming shortwave radiation, as discussed in the next section.





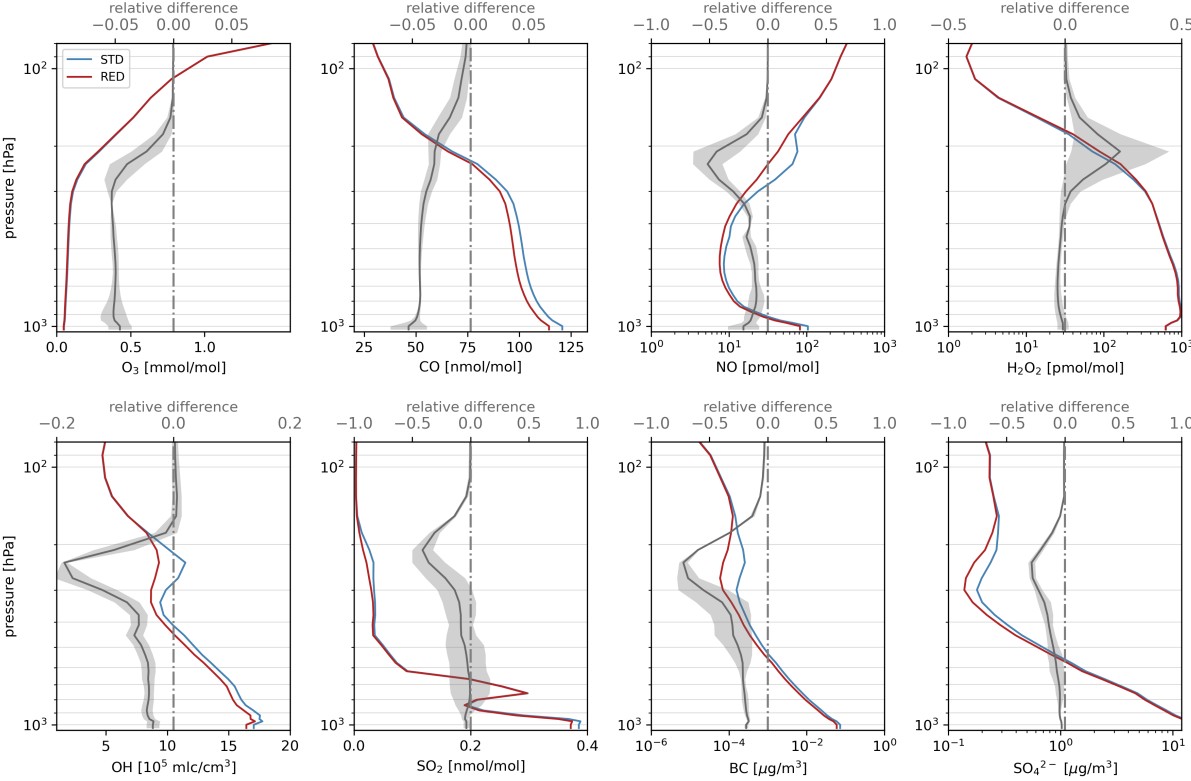

**Figure 5.** Vertical profiles from STD and RED simulations and their relative difference ((RED−STD)/STD). The grey area represents one standard deviation of the spatial-temporal mean (grey line). Please note the different scales for the relative differences.

## 4.2 Impact on radiation

As the model is nudged (i.e. constrained air temperature with prescribed sea surface temperatures), we do not investigate any
effects on the outgoing longwave radiation. Rather, we focus the analyses on the shortwave flux $F_{\mathrm{SW}}$ and its induced heating rate $(\partial T/\partial t)_{\mathrm{SW}}$ in the area encompassing Europe.

### 4.2.1 Direct effects

Aerosols directly impact the radiation balance by absorption and scattering of electromagnetic waves. Compared to the business as usual emissions, the monthly mean sulfate (and inorganic aerosols, not shown) and black carbon concentrations are reduced
most strongly close to the surface, with another (much smaller) local maximum close to the commercial flight level (around 200 hPa, see also Fig. 6). Furthermore, the mean aerosol (number) concentrations were reduced in the lockdown scenario throughout the whole air column (see Fig. 8 and Sect. 4.2.2), with the reduction being most pronounced between 300 and 200 hPa, likely due to reduced aircraft emissions and the effect on particle formation and coagulation at these altitudes.





We calculate the simulated difference in the downwelling shortwave flux between simulation RED and STD, i.e. the impact
of the reduced emissions on the SW radiation. Here only the aerosol contribution is estimated, removing any radiative effect
from changes in trace gases (e.g. ozone) within the Europe longitude-latitude box for May 2020. The differences are largest
over continental central Europe and lowest over Northern Scandinavia (Fig. 7), with no large spatial gradients over Europe.
In virtually all regions there is more downwelling shortwave radiation in the reduced emission scenario. Spatially averaged
at ground level within the European domain, there is an increase of $0.327 \pm 0.105 \, \mathrm{Wm^{-2}}$ under clear sky condition (i.e., no
clouds) compared to the baseline scenario, while at the TOA the increase is $0.198 \pm 0.092 \, \mathrm{Wm^{-2}}$. This increase, together with
the reduced heating rates of ambient air, is indicative of a reduction in shortwave scattering and absorption, due to the reduced
inorganic aerosol and black carbon concentrations, i.e. the lockdown contributed to make the atmosphere more transparent to
SW radiation. We also compared our results with Van Heerwaarden et al. (2021): our radiative effect of all aerosols in our
RED simulation for May is of $-3.33 \pm 1.36 \, \mathrm{Wm^{-2}}$, which is close to their value of $-2.3 \, \mathrm{Wm^{-2}}$. The column integrated
contribution of backscatter and absorption can be estimated from the radiation values at TOA and surface, indicating that,
during lockdown, the total backscatter (clear sky) of SW radiation has been decreased by $0.263 \pm 0.070 \, \mathrm{Wm^{-2}}$, while the
total absorption (clear sky) was decreased by $0.064 \pm 0.053 \, \mathrm{Wm^{-2}}$, with slightly more than one third of this caused by the
BC decrease. Reduced scattering by aerosol particles plays a larger role, as the "net" (i.e. the difference attributable to the
lockdown) shortwave flux is positive in the whole air column; on the other hand, reduced absorption dominates the shortwave
component of direct aerosol effects in the boundary layer, as clearly shown in Fig. 6. The heating of ambient air exhibits a
local minimum in the upper troposphere, which is however small compared to that in the lower troposphere. We calculate the
surface integral of the accumulated heating due to shortwave fluxes, only attributable to aerosols under clear sky conditions:
the difference in the atmospheric layer directly above the surface is $-0.005 \pm 0.001 \, \mathrm{K/day}$, i.e. less heating of the boundary
layer in the lockdown conditions compared to normal emissions. The decreased heating (for the entire column but mostly
at the surface) is due to the reduced absorption by BC during the lockdown conditions, causing a cooling of the atmosphere
(through SW radiation) despite an increase of the incoming radiation. Both the changes in heating and shortwave flux are solely
attributable to the different aerosol burden in the STD and RED simulations.

### 4.2.2 Aerosol–cloud interactions

In Fig. 8, the vertical distributions of the aerosol number concentration (N), ice crystal number concentration (ICNC), cloud
droplet number concentration (CDNC) and ice crystal radius ($r$) are shown for Europe for both simulations, STDCLOUD
and REDCLOUD. Additionally, the SW flux at the TOA and the surface have been calculated from these coupled aerosol–
cloud simulations (see Table 2), both for the total effect (i.e. direct plus indirect) and for the indirect (i.e. neglecting any
direct radiation influence of the aerosol particles). Due to the short simulation period, the difference between these simulations
is much smaller than its variability, represented by its spatial and temporal standard deviation. Nevertheless, comparing the
vertical distribution of number concentrations of aerosols, ice crystals and cloud droplets, the largest relative difference between
STDCLOUD and REDCLOUD (i.e. the two simulations where the aerosol–cloud feedback is activated) is found for the aerosol
number concentration between 200 and 300 hPa. These are the cruise altitudes at which the largest aircraft emissions are





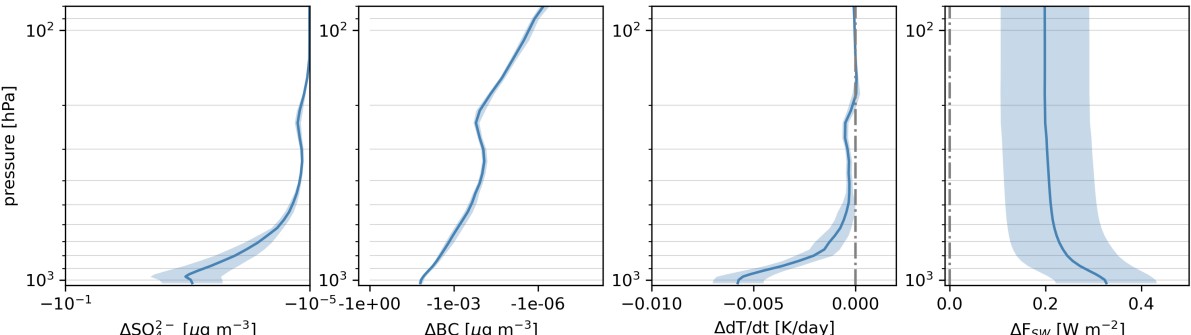

**Figure 6.** Vertical profiles of the difference of monthly mean sulfate mass concentration ($SO_4^{2-}$), black carbon mass concentration (BC), heating rate ($dT/dt$) and the net shortwave flux ($F_{SW}$) between the reduced emission scenario RED and the standard emission scenario STD. shortwave flux and shortwave heating are derived under clear sky conditions. The shading indicates one standard deviation of the monthly mean difference. Note the logarithmic horizontal axis for the two plots on the left.

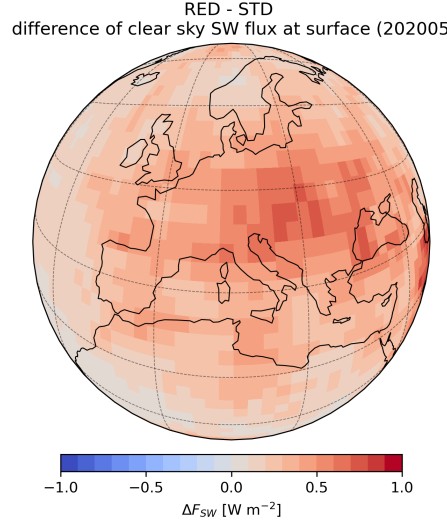

**Figure 7.** Difference of monthly mean clear sky shortwave radiation (May 2020) at the surface between RED and STD simulation. Positive (red) values indicate more incoming radiation at the surface due to less absorption and backscattering in the "lockdown" atmosphere than in the business as usual scenario. Note that we used a common reduction factor for emissions from countries outside Europe.

injected in the model and therefore these differences can be directly connected to the reduced air traffic present during the lockdown (REDCLOUD). As this altitude is somewhat higher than the typical (cold) cloud altitude, the effect on clouds is less
pronounced. At the highest level of these clouds (see Fig. 8) the ICNC are reduced (by $\simeq 30\%$ at 250 hPa, although with large



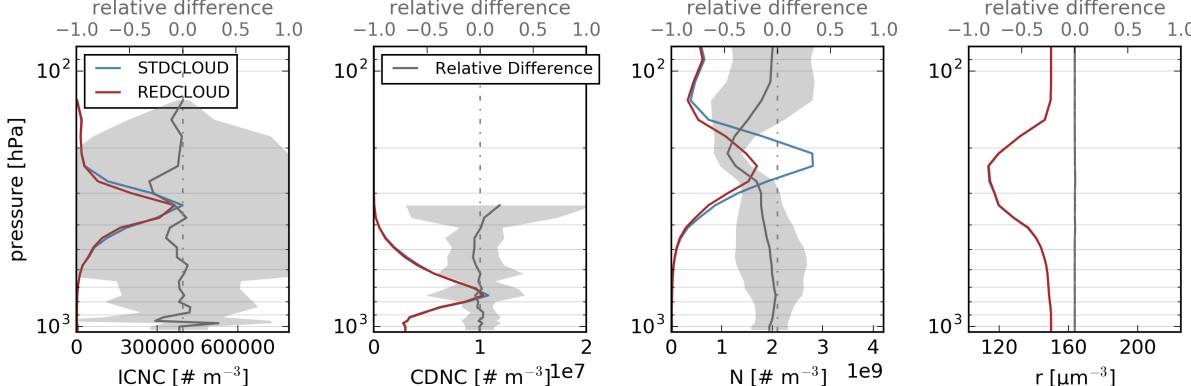

**Figure 8.** Vertical profiles of the monthly mean ice crystal number concentration (ICNC), cloud droplet number concentration (CDNC), aerosol number concentration ($N$) and ice crystal effective radius ($r$) of the reduced emission scenario REDCLOUD (red) and the standard emission scenario STDCLOUD (blue) and their relative difference (grey line) for May 2020 over Europe. The grey area denote the spatial and temporal standard deviation of the relative difference.

variability), while no visible effect is found for CDNC. These results are in line with those obtained by Righi et al. (2021), who showed that aircraft emissions do increase ice crystal number concentration, although their results were not statistically significant. The ice crystal effective radius seems to be the least affected by the reduced emissions during the COVID–19 lockdown, with a negligible absolute and relative difference.

To investigate the effect of reduced aircraft emissions on the SW flux via the indirect aerosol effect at the TOA and surface (SRF), the mean differences in SW flux between REDCLOUD and STDCLOUD for May were calculated over Europe. Positive values indicate greater reflection of SW radiation back to space (for TOA) or more absorption through the troposphere (for surface values) in the STDCLOUD simulation, compared to the REDCLOUD. The mean surface differences are $0.307 \pm 0.115$ Wm$^{-2}$ for the clear sky and $0.443 \pm 1.063$ Wm$^{-2}$ for the all sky case. At the TOA the mean differences in shortwave fluxes are

$0.186 \pm 0.106$ Wm$^{-2}$ (clear sky) and $0.281 \pm 0.928$ Wm$^{-2}$ (all sky, Table 2). We should notice that the clear sky results agree with the direct effect estimated in Sect.4.2.1 but with different simulations, confirming the consistency of the calculations. Thus, the indirect effect of aerosols enhances the direct effect on the SW radiation during the lockdown, even with larger intensity. This confirms the importance of the cloud–aerosol interaction, as mentioned by Hong et al. (2016), Gasparini and Lohmann (2016) and Myhre et al. (2013). However, those values are associated with large standard deviations, related to the

strong spatial variability of the upward shortwave radiation difference between the simulations.

## 5    Conclusions

We simulated the effects of drastically reduced anthropogenic emissions on the atmospheric composition in Europe during the COVID–19 lockdown in spring 2020. We evaluated the model simulations with observations obtained during the aircraft





**Table 2.** Aerosol direct and indirect effects on the shortwave radiation flux at the top of atmosphere (TOA) and surface (SRF) over Europe for May. Note that direct effects are derived from STD and RED simulations, and indirect and total (i.e. direct plus indirect) effects from STDCLOUD and REDCLOUD. The indirect effect of clear sky estimation is obviously equal to zero, but it was included to confirm the validity of the calculations.

| $\Delta F_{SW}$ [Wm$^{-2}$] | RED$-$STD | REDCLOUD$-$STDCLOUD | |
| --- | --- | --- | --- |
| | direct | indirect | total |
| TOA | $0.090 \pm 0.035$ | $0.188 \pm 0.759$ | $0.281 \pm 0.928$ |
| TOA clear sky | $0.198 \pm 0.092$ | $0.000 \pm 0.006$ | $0.186 \pm 0.106$ |
| SRF | $0.209 \pm 0.053$ | $0.233 \pm 1.089$ | $0.443 \pm 1.063$ |
| SRF clear sky | $0.327 \pm 0.105$ | $0.001 \pm 0.023$ | $0.307 \pm 0.115$ |

measurement campaign BLUESKY. The overall agreement between observations and simulated aerosol concentrations and

trace gas mixing ratios is reasonable. Nevertheless, problems remain regarding stratosphere–troposphere transport, especially of volcanic influence, which resulted in systematically underestimated $SO_2$ and $SO_4^{2-}$ of stratospheric origin, and a consequent overestimation of $NO_3^-$ (which substitutes the underestimated sulfate in ammonium salts) in the upper troposphere.

Focusing on the effects of aerosol particles on the shortwave radiation budget, we find that their reduction due to lockdown leads to a net clear sky SW flux increase of $0.327 \pm 0.105$ and $0.198 \pm 0.092$ Wm$^{-2}$ at surface level and TOA over Europe,

respectively. The increase of the SW radiation during the lockdown period is due to the decrease of both black carbon and inorganic aerosols, which made the atmosphere more transparent to the incoming solar radiation by reducing SW absorption and SW backscatter, with the latter dominating. It must be stressed that although this BC reduction causes an increase in the SW incoming radiation, the SW heating has also been reduced by up to $0.005$ K/day, due to the lowered BC absorption.

With reduced emissions, the model simulates a lower number concentration of aerosols; this reduction is located at an

altitude too high to effectively influence the cold cirrus clouds (aircraft cruising altitude, $\simeq 250$ hPa). The analysis of the indirect aerosol effect did not give any conclusive results, due to the large variability in the calculations caused by the short duration of the lockdown "experiment".

Note that contrails and their contribution to radiative forcing are not considered in this study. Contrails are expected to reduce solar radiation reaching the Earth surface and to reduce outgoing longwave radiation. The mean changes induced by reduced

air traffic in 2020 compared to 2019, computed in two model studies, were of the order of $-0.1$ to $0.5$ Wm$^{-2}$ over Europe (Gettelman et al., 2021; Schumann et al., 2021b), so at a magnitude comparable to the aerosol effects found in this study. We therefore plan to include contrail effects in a forthcoming study.

*Code availability.* The Modular Earth Submodel System (MESSy) is continuously further developed and applied by a consortium of institutions. The usage of MESSy and access to the source code is licensed to all affiliates of institutions which are members of the MESSy



Consortium. Institutions can become a member of the MESSy Consortium by signing the MESSy Memorandum of Understanding. More information can be found on the MESSy Consortium Website (http://www.messy-interface.org). The code presented here has been based on MESSy version 2.55 and is available as git commit #dcdc3ed8, in the MESSy repository.

*Data availability.* Data are available from the contact author under request.

*Author contributions.* A.P. and S.R. planned the research. A.P. and S.R. collected and prepared the emission data. A.M. implemented code
corrections for aerosol–cloud interactions. A.P. performed the model simulations. P.J. contributed to the overall model development and helped with the preparation of the model setups. M.K. provided the script for the aerosol mass estimation in the model. Z.H., I.T., L.R., D.J.C. and H.F. provided the data for CO, NO and $H_2O_2$. J.S. and K.K. provided observational aerosol composition data. R.D. and J.N.C. were responsible for the PAN measurements. C.V., L.T. and A.M. provided observational data of $HNO_3$ and $SO_2$. A.Z. provided the ozone data. O.K., B.H., C.P., M.P. and U.P. were conducting, analyzing and interpreting the BC data. M.D. organized the field campaign logistically.
J.C. planned the flight tracks during the campaign. H.S. coordinated the measurements on the FALCON. S.R. and A.P. performed the model evaluation and analysis of direct effects. A.M. and A.P. performed the analysis of indirect effects. U.S. and A.P. discussed the results on the radiative forcing. S.R., A.M and A.P. wrote the manuscript with the help of J.C., M.K. and J.W.. A.P. and J.L. supervised the project. All authors discussed the results and contributed to the review and editing of the manuscript.

*Competing interests.* The authors declare that they have no conflict of interest.

*Acknowledgements.* Christiane Voigt, Laura Tomsche and Andreas Marsing thank funding by the HGF and by the DFG within the SPP1294 HALO under contract no Vo1504/7-1 and the TRR301-1 TP change.



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
