# Peer review of "Numerical simulation of the impact of COVID-19 lockdown on tropospheric composition and aerosol radiative forcing in Europe"

_Atmospheric Chemistry and Physics, 2021_

## Author Comment (AC1)

We thank the reviewer for her/his comments. Here they are repeated (in bold) with our replies.

**The title of this manuscript suggests the authors intend to explain aerosol radiative effects during the early stages of the pandemic with reduced emissions. However, it is not clear to me that they have made a convincing case to justify the title after reading the manuscript. In particular, in the manuscript, they describe a model (nudged by meteorology) with four simulation runs targeting the period of interest. They say their model does a "reasonable" job in general, but this assertion needs further contextualization and reasoning. Overall, a reader like me is left wondering what the authors are actually trying to tell in this manuscript: Is it the "unique" measurement campaign, a "unique" model they developed to match these measurements, or fundamental new progress and understanding about aerosol radiative effects?**

A chemical-climate model of such complexity as the one used our study cannot perfectly simulate observations, not even in specified dynamics mode. On the other hand, a chemistry transport model or air quality model, which might be better suited to simulate the observations, cannot easily be used for radiative impact calculations. We use a CCM (instead of a CTM) to base an estimate of radiative impact (of reduced emissions during the World-wide COVID lock-down) on observations of atmospheric trace gases and aerosols. The agreement between our model simulations and the observations from a tailored measurement campaign using research aircraft is comparably good for most of the trace gases and aerosols, especially when compared to other intercomparison based on other model setup (e.g. ?Pozzer et al., 2022). We believe that our evaluation is quite comprehensive, although room for further analysis is obviously present. In order to avoid giving the false impression, we agree to change the title of the article, to fit better the content of the manuscript. The new title "Numerical simulation of the impact of COVID–19 lockdown on tropospheric composition and aerosol radiative forcing in Europe" reflects the content of the manuscript, for which we simulated the "unique" measurement campaign during the COVID-19 lockdown.

**Major points**

- **What is the purpose of this paper? Is it the model only or the associated knowledge/insight produced by this model in concert with the measurements? There is a clear disconnect between the title and introduction on the one hand and the rest of the manuscript on the other. As an example, there is a lengthy discussion of aerosol effects in the introduction, yet it is not clear how anything in the rest of the paper fills any of the many gaps in our understanding of aerosol–cloud interactions. The authors should consider shortening the introduction (pointing the reader to further material) and instead focus on what they actually address.**

  We thank the referee for this suggestion. We have reduced the introduction. In addition, as the referee pointed out, we decide to change the title of the manuscript, to better reflect the content of the manuscript: "Numerical simulation of the impact of COVID–19 lockdown on tropospheric composition and aerosol radiative forcing in Europe". To our knowledge, no studies on European lockdown have been performed for the entire troposphere, by evaluating model predictions with the observation in the entire tropospheric column. Following the comments of the referee #2, we have instead extended the model description to provide more detailed information on the methodology.

- **The model evaluation is incomplete at best, perhaps quite weak. It can also be circular at times. The authors use a nudging technique whereby they anchor**

**the model to some meteorology, then they evaluate the model by comparing some model outputs to aforementioned meteorology. Is that an accurate reading? Shouldn't they be the same by definition? More context and thorough explanation is needed here.**

This is a classical misunderstanding, at least partly because we did not provide sufficient information about our model setup: we apply a CCM in "specified dynamics" ("nudged") mode, which is established by Newtonian relaxation of the prognostic variables divergence, vorticity, temperature (without global mean), and logarithm of surface pressure. This "nudging" is applied in spectral space by low normal mode insertion, i.e. on the synoptic scale only, in order to nudge the meteorology of the CCM towards the "observed" (indeed analysed or re-analysed) meteorology. Due to the scale limit of this procedure, a CCM in "specified dynamics" mode is still fundamentally different from a CTM, since the nudged CCM develops its own physics on sub-synoptic scales. Therefore, it is required and reasonable to compare the simulated meteorology with the available in-situ observations despite the nudging, as described in line 173. For an in-depth discussion on the Newtonian relaxation technique applied in spectral space, we refer to Jeuken et al. (1996) and Löffler et al. (2016). In our study we show that the agreement between the nudged physical state of the model atmosphere and the observations is good, despite some deviation on specific humidity (a quantity that is NOT nudged). We will clarify this point in the revised version.

- **In general, aerosol indirect effects are challenging and any work purporting to make progress in this field should be scrutinized. So please be precise and forthcoming about what this work actually brings to this field. Again, it is important and timely; so this is not to dismiss this work, but please be as precise as you could to contextualize your work.**

  Indeed, our study does not focus on conceptual advancements or the development of new methodologies. We rather apply a state-of-the-art model to reproduce (by numerical simulation) data observed during the BLUESKY campaign, to further base a qualified estimate of radiative impact (namely of the reduced emissions). We believe our model results to be particularly robust, as the comparison with the observations shows a very good agreement at lower levels, where the highest concentration of aerosols is present. As mentioned above, for clarification, we decided to change the title of the manuscript to focus more on the impact of the world-wide COVID–19 lockdown in Europe.

**Minor comments**

- **While you are free to make up your own definitions, it is often not a good idea to make up acronyms that have other meanings in popular culture or other fields. For example, "STD" refers to sexually transmitted diseases in general, and in this manuscript, it refers to "standard" or "business as usual" — the authors should consider unifying their approach here: They use "STD," "business as usual," and "baseline" throughout the manuscript. One would suffice, preferably the last one, "baseline."**

  Thanks for pointing this out. We changed STD to BASE.

- **In many places there are added sentences that add no value to the text. For example, line 172 could be deleted; the first part of line 189 could also be deleted;**

**lines 105–108 are unnecessary; the majority of line 157 can go as well. More on these in the list of technical comments below.**

We have followed some of the suggestions of the referee (e.g., line 105–108). In other cases (e.g. line 172), we decided to keep the sentence as this is needed to reference the figures and the table.

**List of technical/specific comments:**

Most of the technical/specific comments have been accepted. A few comments deserving a more detailed answer are presented here below.

**Also doesn't "aerosol particle number concentrations" cover the previous parts (e.g. ORG)?**

The ORG refers only to the organic material present in the aerosols (as mass).

**What was the time step of the model? Sampling at 5 minutes seems too frequent for these types of models. Did you simply interpolate from the model time step or what is going on here?**

The model set-up used in our work has a time-step of exactly 5 minutes. We added this information on the model description.

**Line 360: Could you reflect on this range a little more? Seems insignificant and uncertain to a casual reader.**

We have extended the text with the following lines: " The differences between these studies, can partly be attributed to the applied methodologies and general difficulties in discriminating anthropogenic effects from interannual variability. Hence, a study which considers contrail and aerosol effects simultaneously, and covers a longer time period, is recommended to better attribute the causes of the observed changes . "

**Code availability: Why list all these details about doing MOU and all that, can you just give the git repository link and tag/commit?**

Unfortunately, due to the contained legacy models, we are not allowed to host parts of the code open source or publicly accessible. Our GitLab repository is not open and a URL would be therefore meaningless. This section is standard and used in the sister journals GMD(D) as mentioned in the license section of the MESSy system (https://www.messy-interface.org/).

**Data availability: Please make your data available, and refrain from "contact the author" stuff. It doesn't seem open...**

The observational data and the model results are available on the HALO (High Altitude Long RAnge research aircraft) database (https://halo-db.pa.op.dlr.de), upon sign of data protocol. This was included in the revised manuscript.

**References**

Jeuken, A., Siegmund, P., Heijboer, L., Feichter, J., and Bengtsson, L.: On the potential of assimilating meteorological analyses in a global climate model for the purpose of model validation, Journal of Geophysical Research: Atmospheres, 101, 16 939–16 950, 1996.

Löffler, M., Brinkop, S., and Jöckel, P.: Impact of major volcanic eruptions on stratospheric water vapour, Atmospheric Chemistry and Physics, 16, 6547–6562, https://doi.org/10.5194/acp-16-6547-2016, 2016.

Pozzer, A., Reifenberg, S. F., Kumar, V., Franco, B., Kohl, M., Taraborrelli, D., Gromov, S., Ehrhart, S., Jöckel, P., Sander, R., Fall, V., Rosanka, S., Karydis, V., Akritidis, D., Emmerichs, T., Crippa, M., Guizzardi, D., Kaiser, J. W., Clarisse, L., Kiendler-Scharr, A., Tost, H., and Tsimpidi, A.: Simulation of organics in the atmosphere: evaluation of EMACv2.54 with the Mainz Organic Mechanism (MOM) coupled to the ORACLE (v1.0) submodel, Geoscientific Model Development, 15, 2673–2710, https://doi.org/10.5194/gmd-15-2673-2022, 2022.

---

## Author Comment (AC2)

We thank the reviewer for her/his comments. Here they are reported (in bold) with our replies. Following the comments of referee #1 and these comments, we have decided to change the title of the manuscript, to fit better with the content of the manuscript: 'Numerical simulation of the impact of COVID-19 lockdown on tropospheric composition and aerosol radiative forcing in Europe'"

**General comments**

• Introduction: The literature review tends to omit primary references and focuses on recent work, with quite a few imprecise descriptions of key processes (that this long list of expert authors could easily address).

Following the comments of referee #1 the introduction has been revised.

• Methods: The description of the model and of the setup of the simulations is insufficient, which affects reproducibility and the ability to interpret the results.

We have followed the suggestion of the referee (see specific comments) to improve this section. The model description is now more precise and detailed.

• Context : manuscript entirely ignores an international model intercomparison project on this very subject, CovidMIP, which has been published months before submission of this manuscript (Jones et al, GRL, 2021). (Disclaimer: I am not involved in CovidMIP.) Clearly, the results of this study should be put into the available context but beyond that, it needs to be clear what additional insights are gained other than the focus on a specific area. The availability of a dedicated aircraft campaign provides ample opportunity to do this but is currently not exploited beyond a baseline evaluation of the model.

We thank the referee for pointing out this very important intercomparison project, which we indeed have overlooked. We have added this reference in the introduction and we have summarized the outcome of this project. Nevertheless, we argue that our study only partially overlaps with CovidMIP, which does not include a comparison with observations. It is thus difficult to judge, whether the series of model results presented in CovidMIP are reflecting the "real" state of the atmosphere during the lockdown. Here, we us a state-of-the-art CCM to reproduce the observed state of the atmosphere (to the extend that is possible) to base a qualified estimate of radiative impact on this (evaluated) model. We also note that a number of manuscripts is in preparation or have been submitted to a special issue in ACP(D) "BLUESKY atmospheric composition measurements by aircraft during the COVID-19 lockdown in spring 2020" (e.g. Nussbaumer et al., 2021; Hamryszczak et al., 2022). These manuscripts will include aspects of modelling and thus additional insights will be presented.

• Analysis : The interpretation of the results tends to be quite speculative and is held back by not tracing the perturbations through the full chain of relevant processes and by a lack of dedicated sensitivity studies necessary to back up some of the interpretation of the results.

We believe that we were not clear enough in our manuscript. More than 15 different sensitivity simulations were performed for our study, while only the basic simulations are presented, so to not overload the reader with information. We are hesitant to add a description and discussion on all these simulations simulations to the main text, as this would only detract from the main focus and make the manuscript less readable. Therefore we prefer to rephrase parts of the manuscript, also following the suggestions of the referee in the specific comments.

**Specific comments**

We sincerely thank the referee for reading the manuscript accurately and pointing out the text which was incorrect or not detailed enough. We believe that the manuscript has largely improved now that these specific comments have been taken into account.

• Where possible, please use primary references. For example, the trade-off between GHG and aerosols was not discovered in 2019 or the dependence of forcing on surface albedo is not something new from the Belloin et al. (2020) paper...

We consider the work of Bellouin et al. (2020) an excellent review on the topic (with detailed references therein) with up-to-date estimates, and we would like to keep the citation. Never-theless, we further added a citation of Shindell et al. (2013). For the GHG-aerosols trade-off we cite the IPCC report, which offers a comprehensive review of the state-of-the-art for this topic (Myhre et al., 2013).

• Line 73: "The cloud albedo effect can be enhanced by the Twomey effect" is very confusing as they tend to be used synonymously. I do not understand what is meant here.

The introduction has been fully rewritten and this sentence removed (see also replies to referee #1).

• Line 79: "The same effect arises in aircraft flight tracks..." claims analogy between ship-tracks (albedo enhancement of existing clouds via Twomey effect or LWP increase) and the formation contrails but this is really not the same.

The introduction has been fully rewritten and this sentence removed (see also replies to referee #1).

• Line 88: Cloud lifetime effect is introduced without giving credit to Albrecht and treated as a fact, rather than a long-standing (and often disputed) hypothesis. The cited references are fairly outdated.

The introduction has been fully rewritten and this sentence removed (see also replies to referee #1).

• Line 132: "The aerosol-cloud interactions are based on the aerosol microphysics parameterization of Pringle et al. (2010) including aerosol aging and the continuous calculation of aerosol number concentration depending on the mass mixing ratio and mixing state." This is not a description of aerosol-cloud interactions but of the underlying aerosol microphysics. Which key processes are represented and how? To name a few: updraft velocities, activation, the link from activated particles to CDNC (in particular in presence of existing droplets), the effect of CDNC on cloud microphysical (through autoconversion/accretion) and radiative properties.

We agree that the sentence was not precise and the scheme of Pringle et al. (2010) indeed describes the aerosol microphysics. We modified the sentence by adding more information on the aerosol representation (we added information on cloud formation and the aerosol-cloud

interactions in the next reply (L135)). The new text is the following: "Atmospheric aerosols are described via a two-moment aerosol scheme, which predicts number concentration and mass mixing ratio of the aerosol modes (Pringle et al., 2010). This scheme takes into account various physical-chemical processes of aerosols, such as coagulation, aging, condensation, and also the gas-aerosol partitioning (Fountoukis and Nenes, 2007)."

- Line 135: "Large-scale cloud formations and prognostic variables depending on cloud microphysical processes follow the work of Lohmann et al. (2007); Lohmann and Hoose (2009); Bacer et al. (2018)." This seems unlikely as Lohmann et al describe a cloud microphysics scheme and you seem to refer to the cloud fraction scheme (which presumably is Sundquist but this is not described at all).
  - The sentence was not precise, because Lohmann et al. indeed describe a two-moment cloud scheme and we did not specify the cloud cover scheme used in the simulations (the one of Sundqvist et al., 1989). We changed this sentence and we wrote more details on the cloud representation and the interplay with aerosols as following: "Convective cloud processes are accounted for using the framework of Tost et al. (2006), based on the convection schemes of Tiedtke (1989) and Nordeng (1994). Convective cloud microphysics does not take into account the influence of aerosols on liquid droplet or ice formation processes and is solely based on temperature and vertical velocity. In EMAC, the vertical velocity is given by the sum of the grid mean vertical velocity and the turbulent contribution (Brinkop and Roeckner, 1995), thus one single updraught velocity is used for the whole grid cell. Large-scale stratiform clouds are described by the CLOUD submodel, which, in the setup applied here, uses a two-moment cloud microphysics scheme for cloud droplets and ice crystals (Lohmann et al., 1999; Lohmann and Kärcher, 2002; Lohmann et al., 2007). and solves the prognostic equations for specific humidity, liquid cloud mixing ratio, ice cloud mixing ratio, cloud droplet number concentration (CDNC), and ice crystal number concentration (ICNC). Cloud droplet formation is computed by the "unified activation framework", an advanced physically based parameterization (Kumar et al., 2009; Karydis et al., 2011) that combines the  $\kappa$ -Köher theory (Petters and Kreidenweis, 2007) for the activation of soluble aerosols with the Frenkel-Halsey-Hill adsorption activation theory (Kumar et al., 2009) for the droplet activation due to water adsorption onto insoluble aerosols. Ice formation occurs via homogeneous ice nucleation following the parameterization of Barahona and Nenes (2009) and heterogeneous ice nucleation of insoluble dust, insoluble black carbon, and glassy organics following Phillips et al. (2013). In the cirrus regime  $(T \leq 238.15K)$ , the effect of pre-existing ice crystals and the competition for the available water vapor between homogeneous and heterogeneous ice nucleation mechanisms are taken into account (Bacer et al., 2018). Given the high contribution of instantaneous freezing (Bacer et al., 2021), the ICNC in the cirrus regime was modified according to Neubauer et al. (2019) in order to reduce the artificial homogeneous freezing of dry aerosol particles independent of availability of water vapor. Other microphysical processes related to cloud droplets and ice crystals, like phase transitions, auto conversion, aggregation, accretion, evaporation, melting, are also taken into account by the CLOUD submodel. The cloud cover is computed diagnostically with the scheme of Sundqvist et al. (1989), which is based on the grid-mean relative humidity."
- Line 140...: "We performed four simulations ... without cloud-aerosol interaction" casually refers to simulations performed without cloud-aerosol interactions. This is not a trivial exercise using a two-moment cloud microphysics as clouds

droplet number concentrations are prognostic and, if decoupled from aerosols, need to be initialised somehow (and the base-state will affect the results due to inherent nonlinearities) but no details are given on how this is done.

We apologize for the information missing in the manuscript. We augmented the manuscript with the following lines: "The model setup without cloud-aerosol interactions uses the original ECHAM5 cloud microphysical scheme (Lohmann and Roeckner, 1996) and a statistical cloud cover scheme including prognostic equations for the distribution moments (Sundqvist et al., 1989). Details on the cloud microphysical scheme can be found in Roeckner et al. (2003, and references therein)."

• It is difficult to put the results from this study into the wider context, such as AeroCom or CovidMIP, without a summary of the of ERFari and ERFaci from PD-PI simulations. As we currently have limited constraints on ERFaci from observations, it is important to know where the model lies in the ERF uncertainty range e.g. from IPCC AR6 or the Bellouin et al (2020) assessment. This should be included and discussed either in the methods or results section.

An analysis of direct and indirect effects with a model set-up very similar to the one used in our study can be found in Lelieveld et al. (2019). With regard to direct and indirect effect (and their spatial distribution) we refer to Fig. 5 of their supplement. We added the following lines in the revised manuscript: "Following the work of Lelieveld et al. (2019), the EMAC model in a setup very similar to ours, simulates a radiative forcing global mean of all anthropogenic aerosols at TOA (top of the atmosphere) of  $-0.46 \pm 0.01Wm^{-2}$  and  $-1.2 \pm 0.1Wm^{-2}$  for ERFari and ERFari+ERFaci, respectively. At BOA (bottom of the atmosphere) the model simulates  $-1.6 \pm 0.02W/m^{-2}$  and  $-2.1 \pm 0.1W/m^{-2}$  for ERFari and ERFari+ERFaci, respectively."

• Line 164: Measurement cut-offs are quoted but it is not clear if and how these are applied to the model size distributions in the evaluation. Are they explicitly applied for each component, how are internal mixtures dealt with – or are they ignored? And if they are, how would this affect the results?

The measurement cut-offs are applied to the model results when these are compared to the observations, in order to compare the same size range. More precisely they are applied for each individual component, to obtain a meaningful comparison for each instrument/component. The aerosols in the model are considered to be internally mixed (see reference to Pringle et al. (2010)). We clarified this point in the revised manuscript.

• Figure 2: The evaluation of O3, CO, NO is looking very good. Has there been any calibration/tuning during the setup of the simulations or is this out of the box?

The model was not tuned for these trace gases. The results are obtained from the simulation results as described in Sect. 2.1.

• Line 206: Here and later it is hypothesized that the underestimation of SO2 (and later on SO4) is due to representation of transport from the boundary layer or from the stratosphere to the upper troposphere or due to model short-comings within the stratospheric aerosol chemistry – but no further sensitivity studies are conducted to underpin this hypothesis.

We understand the concern of the referee and we agree that this should be investigated in more detail. As both, sulfate and  $SO_2$ , are simultaneously underestimated, this points to some model deficit, and in the manuscript, we indicate that this is most probably (due to strong sulfate and ozone correlation) of stratospheric origin (e.g. due to volcanic eruption, line 243). In fact, at the altitudes where sulfate was measured, ozone has practically only stratospheric origin, and the strong correlation implies same sources for the sulfate. Nevertheless, despite various sensitivity simulations with the inclusion of the various eruptions prior to the field campaign, the underestimation is still present. Therefore, a more comprehensive study is needed to fully understand the model deficiencies, which is ongoing. Although the sulfur dioxide emissions could be increased to match the observational data, this would not help to solve the real problem. More detailed work, with additional observational datasets (e.g. satellite observations, field campaigns at similar and higher altitudes) is necessary to fully investigate this issue, and this is clearly outside the scope of our study. In addition, also instrumentation issues at higher altitudes cannot be ruled out: in fact, on a recent comparison with the same model and identical emissions total, the sulfate profiles compared to measurements from various aircraft campaigns were quite satisfactory (Pozzer et al., 2022).

• Line 217: "The measured black carbon (BC) concentrations are captured well by the model close to the surface, while the observational variability is underestimated at high altitudes." This seems to neglect the significant bias – it looks like median concentrations are almost an order of magnitude out in the upper troposphere? This section also needs to explicitly caution (not only in the caption) that you switched plots from a linear to a log scale

Indeed the referee is correct, with the observed median being much lower than the simulated one. However, it must be stressed that the measurement variability is extremely large, showing a very skewed distribution of the measurements at such altitudes. A detailed analysis of the black carbon concentration simulated with an almost identical model setup can be found in Krüger et al. (2022), and therefore we prefer not go into detail in our manuscript. We added to the these informations to the manuscript.

• Line 235: "A single factor causing the model underestimation of BC and sulfate aerosol concentrations in the upper troposphere, e.g. a localized plume of pollution, is judged unlikely, as BC and SO2 do not correlate" I am not sure I follow the logic here. This would be true if both would stem from the same source but for plumes arising from entirely different sources it seems plausible to find low correlations – while biases may still be affected by the same process such as a common transport or removal process.

The referee is completely right. In fact, the sentence should read: "A single emission source causing the model underestimation of BC and sulfate aerosol concentrations in the upper troposphere, is judged unlikely, as BC and  $SO_4^{2-}$  do not correlate".

• Line 269: "It must be stressed however, that those relative changes in 270 the upper troposphere, although significant, have a very minor impact on most trace gas budgets, due to their low mixing ratios at these altitudes." Mixing ratio is conserved under vertical displacement – you probably mean low concentrations (due to exponential pressure decrease)?

We have removed the sentence for clarity.

• Figure 3 5: are concentrations normalized to STP (needs to be clear in the caption)?

The concentration are not normalized to STP. We have add this information to the manuscript.

• I am missing an effort to use interesting measurement data and the evaluation to provide some constraint or context for the following analysis of aerosol radiative effects. As a minimum it would be helpful to analyse if the simulated change in response to the emission perturbations are larger than the underlying model biases (which would add trust) or not (which would add less trust).

We thank the referee for pointing this out. Although two sections are dedicated to the evaluation (3.1 and 3.2) and one (4.1) to the change in mixing ratios/concentrations, we did not spell out if "the simulated change in response to the emission perturbations are larger than the underlying model biases". From a detail analysis, the perturbation observed from the COVID-19 lockdown is, for the tracer, generally larger than the model bias to the observations. On the other side, for the aerosols components, the perturbation is smaller than the bias observed (see Figs.2,3 and 5). Nevertheless, we argue that the bias in the comparison would be present in all model simulations, therefore canceling out once the effects on radiation are estimated by subtracting the model results. To make the reader however aware of the issue, the following text has been added to the revised manuscript: "While the changes in CO and NO can be considered significant and representative of the real atmospheric changes (due to the low bias at all tropospheric levels between model results and observations), changes in the aerosol components should be considered with caution, as these are generally smaller than the bias between the model results and the observations."

• This section (and subsequent use) should stick to well defined nomenclature of aerosol forcing as used by IPCC, i.e., be clear what is RF, what is ERF, what is ari and what is aci. This also means that ERF should include SW and LW and it is not clear why the analysis is restricted to SW only.

Indeed the referee is here completely right, and we apologize for having ignored the IPCC nomenclature. First of all, the forcings included in our study are, following the Myhre et al. (2013) definitions, RF (radiative forcing) and not IRF (Instantaneous Radiative Forcing) or ERF (Effective Radiative Forcing). This is due to the fact that the tropospheric temperature is somewhat constrained by the prescribed SST/SIC (consistent with the nudging data) and the temperature nudging (patterns only), while the stratosphere is dynamically free. Moreover, for the applied model setup the "meteorology" of the BASE and RED simulations are binary identical. It must be stressed that the EMAC setup used here was in Quasi-Chemistry Transport Model (Q-CTM Deckert et al., 2011), and therefore we cannot derive IRF or ERF as in a fully coupled model setup. We have clarified this in the manuscript. Furthermore, the RF estimated are the effect of the COVD-19 lockdown against a baseline scenario (i.e. Business as usual), and therefore does not represent the radiative effect of all anthropogenic aerosols.

For the sake of completeness, we have also added the  $RF_{aci}$  and  $RF_{ari}$  to the manuscript (see Tab.1), although we decided to keep the analysis of the SW radiation as it is, as, to our knowledge, these are the most meaningful results, as the LW radiation is constrained due to the prescribed SST/SIC and the temperature nudging. Furthermore, following the definitions in Myhre et al. (2013) and Bellouin et al. (2020),  $RF_{ari}$  and  $RF_{aci}$  cover both

Table 1:  $RF_{ari}$  and  $RF_{aci}$  at the top of atmosphere (TOA) over Europe for May for the lockdown against a baseline scenario.

|     | $\mathrm{RF}_{\mathrm{ari}}$ | $\mathrm{RF}_{\mathrm{aci}}$ |
|-----|------------------------------|------------------------------|
| TOA | $0.08\pm0.03$                | $0.19\pm0.92$                |

the solar (shortwave, SW) and terrestrial (longwave, LW) parts of the electromagnetic spectrum. Therefore we decided to not use this nomenclature when writing only of the shortwave radiation, so to avoid possible misunderstanding.

We would like also to point out that even in the CovidMIP analysis performed by Jones et al. (2021), only the shortwave radiation is investigated (aerosol radiation interaction), without any reference to total radiation or aerosol cloud interactions, as this was the only statistically significant anomaly identified. Analogously to us, Jones et al. (2021) also avoided the nomenclature  $RF_{aci}$  and  $RF_{ari}$ , as only the shortwave radiation were taken into account (see above).

• This section would be much more intuitive if it followed the actual chain of processes from aerosol properties, through aerosol radiative properties (AOD, AAOD) all the way to the radiative fluxes.

We thank the referee for the suggestion. Nevertheless, we prefer to keep the structure as it is and extend as possible the missing part, so to facilitate a comparison with the original manuscript.

**• Line 289: Fluxes defined at what level?**

This is mentioned in line 294: ground level.

• Line 298: "our radiative effect of all aerosols in our RED simulation for May is of  $3.33 \pm 1.36$  Wm2, which is close to their value of 2.3 Wm2." The definition of "radiative effect" is entirely unclear here.

The referee is right. The sentence is not clear and based on erroneous data. We have removed the sentence as it was not essential for the manuscript.

• Line 303: "the total absorption (clear sky) was decreased by  $0.064\pm0.053$  Wm2, with slightly more than one third of this caused by the BC decrease" How do you know?

As mentioned before, various sensitivity studies were performed to proof our assertions. In this case we performed a simulation equal to STD (now BASE), but only with decreased BC emissions according to the lockdown. Similar simulations were performed performed for  $SO_2$  and NO, to analyze the radiative effects of reducing individual components. Although the individual perturbation results are not summing up linearly, the radiation impact of the individual reduction during lockdown provided a fully closed budget within their uncertainty ranges. We added to the revised manuscript the text in line 303: "was decreased by  $0.06 \pm$  $0.05Wm^{-2}$ . Based on an additional sensitivity simulations, in which only individual emissions (i.e. of BC,  $SO_2$ , NO) have been reduced, we found that slightly more than one third of this reduction is caused by the BC decrease".

Figure 1: Simulated differences in heating rates with a reduction of all emissions (black) and a reduction of BC emissions only (red).

• Line 309: "The decreased heating (for the entire column but mostly 310 at the surface) is due to the reduced absorption by BC". Again, how do you know? We thank the referee for pointing this out. As mentioned in the previous answer, this is based on the results of additional sensitivity simulations to evaluate the importance of BC. We checked again the results and, as shown in Fig.1, we found that the decrease of BC is responsible for roughly 40% of the decrease of the shortwave heating at the surface. We have corrected this in the revised manuscript.

**• Line 314 / Fig 8. What is N – how is it defined? Is it total CN without size-cut off?**

In contrast to the comparison with the observations (which had a cut off), N is here the total aerosol number concentration simulated by the model. Analogously, also ICNC and CDNC are without size cut off. We added this information to the revised manuscript.

• Line 323: "these differences (in N, CDNC, ICNC) can be directly connected to the reduced air traffic present during the lockdown (REDCLOUD)" UTLS aerosol tends to be dominated by nucleation (not sure if this included in N or not as it is not defined) so the attribution to aircraft is ambiguous. I am really missing a process-based analysis here from emission to CN to CCN/INP to CDNC/ICNC – and from the model description it is not even clear what processes could actually affect CDNC/ICNC. Likewise, the attribution to specific emission sectors should not be based on speculation – it would be trivial to run a simulation with and without the aircraft only emission reductions to make this point.

As part of the sensitivity simulations performed, we have performed a simulation, in which the BASECLOUD (formerly STDCLOUD simulation) was repeated, but with reduced aircraft emissions. In Fig. 2 the results are shown, compared to the simulation BASECLOUD. It is shown that the reduction of aircraft emissions caused a strong decrease in the particle

Figure 2: Vertical profiles of the monthly mean aerosol number concentration from simulation BASECLOUD (blue), REDCLOUD(RED) and a sensitivity simulation based on the same but with only reduced aircraft emissions (green) for May 2020 over Europe. The grey line depicts the relative difference between REDLCOUD and the sensitivity simulation. The grey area denotes the spatial and temporal standard deviation of the relative differences.

numbers in the upper troposphere. We calculated, from our model results, that more than 90% of the reduction of particle number concentrations at 200-300 hPa are attributable to the missing aircraft emissions at these altitude, while less than 10% are caused by reduction of other emissions, proving that aircraft emissions reduction is mostly responsible for the strong decrease in particle number at 200-300 hPa.

• Line 334: There is really very limited point to quote RF/ERF with three significant figures in the presence of significant noise and uncertainty.

We agree with the referee and we modified the figures to two significant digits.

• Line 338: "This confirms the importance of the cloud-aerosol interaction, as mentioned by Hong et al. (2016), Gasparini and Lohmann (2016) and Myhre et al. (2013)." This is of course not new but the split is highly model dependent so this should be discussed early on (as suggested above).

We have removed this line and included this discussion in the introduction.

• Line 345: "Nevertheless, problems remain regarding stratosphere-troposphere transport, especially of volcanic influence, which resulted in systematically underestimated SO and SO2of stratospheric origin, and a consequent overestimation of NO3 (which substitutes the underestimated sulfate in ammonium salts) in the upper troposphere." This could be true but no results are provided to underpin this, nor is a reference given that shows this.

This has been proven in many studies and is of common knowledge (e.g., Seinfeld and Pandis, 2008; Bauer et al., 2007; Xu and Penner, 2012). These reference were added to the revised manuscript.

• Line 354: "With reduced emissions, the model simulates a lower number concentration of aerosols; this reduction is located at an altitude too high to effectively influence the cold cirrus clouds" From the model description it is entirely unclear if or how "aerosols" could actually affect cirrus in this setup.

We included information about the influence of aerosols on cloud formation in the model description (section 2.1, see replies to L132 and L135) and we changed the sentence at L354 to: "With reduced emissions, the model simulates a lower number concentration of aerosols between 300 and 50 hPa; this reduction is located at an altitude too high to influence the cloud droplet formation (Karydis et al., 2017) and heterogeneous ice nucleation from black carbon and dust; glassy organics freeze at these altitudes, but their contribution is totally negligible in comparison with homogeneous nucleation (Bacer et al., 2021)."

• Line 356: "The analysis of the indirect aerosol effect did not give any conclusive results, due to the large variability in the calculations caused by the short duration of the lockdown "experiment". I agree but you also make the argument that these "effects" dominate the overall result, so this suggests that this could be noise?

We partially disagree with the referee: these effect dominates only for the SW radiation at the TOA. At the surface, the aerosol cloud interaction effect on SW radiation is comparable to the direct effect, although with much larger variability. On the other hand, the referee is right in mentioning that the changes of cloud properties add noise to the system, which masks the overall effect. Possibly, an ensemble of simulations (or a much longer simulation) could reduce the variability, although this is outside the scope of our study.

• At a minimum, the data going into the plots should be deposited in an open access archive.

The observational data and the model results are available on the HALO (High Altitude Long RAnge research aircraft) database (https://halo-db.pa.op.dlr.de), upon sign of data protocol. This was included in the revised manuscript.

[revised manuscript text omitted]

---

## Author Response (AR2)

Dear editor, here a item-by-item response of referee #1 comments. As we believe only referee #1 specific points where missing, we have added only those to this document. We listed each single change with also the line numbers of the new manuscript.

For safety we attached also the differences between the updated manuscript and the on-line version at the end of this document.

Best regards,

Andrea Pozzer on behalf of all authors.

**The title of this manuscript suggests the authors intend to explain aerosol radiative effects during the early stages of the pandemic with reduced emissions. However, it is not clear to me that they have made a convincing case to justify the title after reading the manuscript. In particular, in the manuscript, they describe a model (nudged by meteorology) with four simulation runs targeting the period of interest. They say their model does a "reasonable" job in general, but this assertion needs further contextualization and reasoning. Overall, a reader like me is left wondering what the authors are actually trying to tell in this manuscript: Is it the "unique" measurement campaign, a "unique" model they developed to match these measurements, or fundamental new progress and understanding about aerosol radiative effects?**

A chemical-climate model of such complexity as the one used our study cannot perfectly simulate observations, not even in specified dynamics mode. On the other hand, a chemistry transport model or air quality model, which might be better suited to simulate the observations, cannot easily be used for radiative impact calculations. We use a CCM (instead of a CTM) to base an estimate of radiative impact (of reduced emissions during the World-wide COVID lock-down) on observations of atmospheric trace gases and aerosols. The agreement between our model simulations and the observations from a tailored measurement campaign using research aircraft is comparably good for most of the trace gases and aerosols, especially when compared to other intercomparison based on other model setup (e.g. Jöckel et al., 2010; Pozzer et al., 2022). We believe that our evaluation is quite comprehensive, although room for further analysis is obviously present. In order to avoid giving the false impression, we agree to change the title of the article, to fit better the content of the manuscript. The new title "Numerical simulation of the impact of COVID–19 lockdown on tropospheric composition and aerosol radiative forcing in Europe" reflects the content of the manuscript, for which we simulated the "unique" measurement campaign during the COVID-19 lockdown.

**Major points**

- **What is the purpose of this paper? Is it the model only or the associated knowledge/insight produced by this model in concert with the measurements? There is a clear disconnect between the title and introduction on the one hand and the rest of the manuscript on the other. As an example, there is a lengthy discussion of aerosol effects in the introduction, yet it is not clear how anything in the rest of the paper fills any of the many gaps in our understanding of aerosol–cloud interactions. The authors should consider shortening the introduction (pointing the reader to further material) and instead focus on what they actually address.**

  We thank the referee for this suggestion. We have reduced the introduction. In addition, as the referee pointed out, we decide to change the title of the manuscript, to better reflect the content of the manuscript: "Numerical simulation of the impact of COVID–19 lockdown on tropospheric composition and aerosol radiative forcing in Europe". To our knowledge, no

studies on European lockdown have been performed for the entire troposphere, by evaluating model predictions with the observation in the entire tropospheric column. Following the comments of the referee #2, we have instead extended the model description to provide more detailed information on the methodology.

- **The model evaluation is incomplete at best, perhaps quite weak. It can also be circular at times. The authors use a nudging technique whereby they anchor the model to some meteorology, then they evaluate the model by comparing some model outputs to aforementioned meteorology. Is that an accurate reading? Shouldn't they be the same by definition? More context and thorough explanation is needed here.**

This is a classical misunderstanding, at least partly because we did not provide sufficient information about our model setup: we apply a CCM in "specified dynamics" ("nudged") mode, which is established by Newtonian relaxation of the prognostic variables divergence, vorticity, temperature (without global mean), and logarithm of surface pressure. This "nudging" is applied in spectral space by low normal mode insertion, i.e. on the synoptic scale only, in order to nudge the meteorology of the CCM towards the "observed" (indeed analysed or re-analysed) meteorology. Due to the scale limit of this procedure, a CCM in "specified dynamics" mode is still fundamentally different from a CTM, since the nudged CCM develops its own physics on sub-synoptic scales. Therefore, it is required and reasonable to compare the simulated meteorology with the available in-situ observations despite the nudging, as described in line 173. For an in-depth discussion on the Newtonian relaxation technique applied in spectral space, we refer to Jeuken et al. (1996) and Löffler et al. (2016). In our study we show that the agreement between the nudged physical state of the model atmosphere and the observations is good, despite some deviation on specific humidity (a quantity that is NOT nudged). We will clarify this point in the revised version.

- **In general, aerosol indirect effects are challenging and any work purporting to make progress in this field should be scrutinized. So please be precise and forthcoming about what this work actually brings to this field. Again, it is important and timely; so this is not to dismiss this work, but please be as precise as you could to contextualize your work.**

Indeed, our study does not focus on conceptual advancements or the development of new methodologies. We rather apply a state-of-the-art model to reproduce (by numerical simulation) data observed during the BLUESKY campaign, to further base a qualified estimate of radiative impact (namely of the reduced emissions). We believe our model results to be particularly robust, as the comparison with the observations shows a very good agreement at lower levels, where the highest concentration of aerosols is present. As mentioned above, for clarification, we decided to change the title of the manuscript to focus more on the impact of the world-wide COVID–19 lockdown in Europe.

**Minor comments**

- **While you are free to make up your own definitions, it is often not a good idea to make up acronyms that have other meanings in popular culture or other fields. For example, "STD" refers to sexually transmitted diseases in general, and in this manuscript, it refers to "standard" or "business as usual" — the authors should**

consider unifying their approach here: They use "STD," "business as usual," and "baseline" throughout the manuscript. One would suffice, preferably the last one, "baseline."

Thanks for pointing this out. We changed STD to BASE.

- **In many places there are added sentences that add no value to the text. For example, line 172 could be deleted; the first part of line 189 could also be deleted; lines 105–108 are unnecessary; the majority of line 157 can go as well. More on these in the list of technical comments below.**

We have followed some of the suggestions of the referee (e.g., line 105–108). In other cases (e.g. line 172), we decided to keep the sentence as this is needed to reference the figures and the table.

**List of technical/specific comments:**

- **COVID-19 takes a dash, not en or em dash.**

We have used only dash in the revised manuscript

- **Line 2: reads awkwardly "through direct... and indirectly," better use rephrase to use "directly ... and indirectly"**

Line has been modified following the suggestion as " Aerosols influence the Earth's energy balance directly by modifying the radiation transfer and indirectly by altering the cloud microphysics."

- **Line 4: delete "Here"**

The "Here" has been deleted

- **Line 9: delete "a somewhat"**

The "a somewhat" has been deleted

- **Line 10: delete "which could have ... campaign"**

The "which could have influenced the campaign" has been deleted

- **Line 10–11: replace "a business as usual scenario" with "the baseline"**

We have replace all the "business as usual scenario" with "baseline" in all the text, i.e. line 6 , line 10, line 35, line 57, line 86, line 152, line 256, line 283, line 299, line 313, line 331, label Figure 7, line 322, label Table 2.

- **Line 16–17: reads unclearly, maybe write: "ice crystal concentration, cloud droplet number concentration, and effect..."**

We have change the sentence following the suggestion and now reads " Impacts on ice crystal concentrations, cloud droplet number concentrations,and effective crystal radii are found to be negligible. "

- **Line 19: "millions of years of life expectancy" — not sure what the cited items say, but last time I checked, life expectancy refers to one person's life expectancy and so summing the whole planet's life expectancies to make a point is both**

**unscientific and clumsy. I will leave it up to you to decide, but hopefully you will decide to keep the convention.**

We have rephrase the sentence (line 2s1921 with) " They have large impact on human health [. . . ]"

- **Line 24: replace "here" with "hereafter"**

  We have replaced "here" with "hereafter"

- **Line 27: by citing many works on air pollution and not citing a single work on climate effects, you're positioning yourself as the only paper addressing this. First, that's wrong because you're not making a convincing case here anyway about climate effects per se. Second, there are many studies about the climate effects of the lockdown. Could you please cite them and contextualize how your work differs from them?**

  We have removed the sentence "and climatic effects, on which we will focus in this work.", as now we have new reference on climate studies, also based on the reply to referee #. In line 36-40 we added the sentence " Many works are present in the literature that investigate the climatic effect of COVID-19 lockdown (e.g. Lee et al., 2021; Forster et al., 2020; Gettelman et al., 2021). Of particular importance is the CovidMIP intercomparison project, where 12 global chemistry climate model were used to investigate the impact of COVID-19 lockdown on the radiation (Jones et al., 2021; Lamboll et al., 2021), with special focus on aerosol-radiation interaction. "

- **Line 35: replace "business as usual" with "baseline"**

  We have replace all the "business as usual scenario" with "baseline" in all the text, i.e. line 6 , line 10, line 35, line 57, line 86, line 152, line 256, line 283, line 299, line 313, line 331, label Figure 7, line 322, label Table 2.

- **Line 37: remove ", as will be ..."**

  The sentence has been removed.

- **Line 43: "wavelength of the radiation" — last time I checked this was more or less constant or basically unchanging during the lockdown, so what gives? Why do you have it here? It seems you're implying that it is changing...**

  The sentence is in our point of view correct: absorption and scattering of aerosols depends on the wavelenght of the radiation (Seinfeld and Pandis, 2008).

- **Line 50: last word, "May" — which May? May 2020?**

  The "2020" has been added in line 54.

- **Line 55: "trigger several indirect effects" — awkward phrasing**

  The "trigger several indirect effects" has been removed and replaced with "have several indirect effects".

- **Line 56: "alter cloud properties" should be better phrased, maybe "can potentially alter cloud properties" or something similar**

  We have replaced "alter cloud properties" with "can potentially alter cloud properties"

- **Line 64: avoid using two symbols after each other, use the word "approximately" maybe.**

  Line 64 has been removed, as part of the introduction has been rewritten based also on referee #2 comments.

- **Line 99: "unique" may be a stretch.**

  The word "unique" has been removed (line 87), and now reads " We use the BLUESKY observational data set of trace gases and aerosols obtained during an aircraft measurement campaign in Europe during the COVID-19 lockdown in summer 2020 to evaluate the model results. "

- **Lines 105–108 should be deleted**

  It has been removed following referee's suggestion.

- **Lines 140–149: please use something other than "STD" here.**

  We changed the name to "BASE".

- **Line 150: Is the "binary identical dynamics" relevant to your case? If so, please say more about it here briefly.**

  We have extended the description with the text "i.e. they reproduce numerically exactly the same dynamics", line 162.

- **Lines 157–158: "led by ... with the aim of" can be deleted, the interested reader can just go read Voigt et al 2021 if they want.**

  The sentence has been removed.

- **Lines 164–165: not sure if 40 micron is correct, is it? Also doesn't "aerosol particle number concentrations" cover the previous parts (e.g. ORG)? So what's going on with this list?**

  The numbers are correct: line 164 is 40 nm, while line 165 are 40 $\mu$m. The ORG refers only to the organic material present in the aerosols (as mass).

- **Line 169: "sampled online" was the model running on the aircraft? If not, this sentence is wrong**

  We have rewritten the sentence substituting "online" with "runtime" to make it clearer: " For the comparison, the model output was sampled during runtime by the submodel S4D (Jöckel et al., 2016), following the flight tracks of the field campaign and with a time frequency of 5 minutes. "

- **Line 170: What was the time step of the model? Sampling at 5 minutes seems too frequent for these types of models. Did you simply interpolate from the model time step or what is going on here?**

  The model set-up used in our work has a time-step of exactly 5 minutes. We added this information on the model description, in line 103 " [. . . ] the model has a time step of 300 seconds [. . . ] ".

- **"Results: Model evaluation"** — **either use an overarching one "Results" section or just drop the word "results" from sections 3 and 4.**

  We have removed the "Results" word, keeping only "Model evaluation".

- **Line 172 can go.**

  The sentence has been removed.

- **Lines 173–178 can also go; you should generally address the model validation better and this paragraph doesn't do you any service.**

  Based on our reply to the "Major Points" above, we hope we clarified the importance of the temperature evaluation. We kept therefore the sentence as it is.

- **Lines 179–181: yes, exactly. So maybe a different evaluation is needed**

  Based on our reply to the "Major Points" above, we hope we clarified why the dynamics was evaluated. We have rephrase the sentence (line 188) as " Overall, the agreement between the meteorological variables from model and the one observed in the BLUESKY campaign indicates successful initialization and nudging of meteorological variables and that the meteorological conditions during the relevant time period are simulated adequately. "

- **Line 182: "is not surprising" to whom? You could be more precise here**

  The sentence has been reformulated (line 192), clarifying why the deviations should be expected: " As the model is not nudged in the stratosphere or boundary layer (the nudging coefficient is maximal in the free troposphere (Jöckel et al., 2006)), the deviation in the upper troposphere between model results and observational data are to be expected, due to the intrinsic model dynamics which deviates from the nudging data. "

- **Line 189: "Observed ... by the model" should go**

  We have rewritten the sentence (lines 199-202) " More than 94 % of simulated ozone ($O_3$) mixing ratios are within a factor of 2 of the observations ("PF2" value) and the normalized root mean squared error of 0.04 is low, with improvements from previous evaluation of the same model (Jöckel et al., 2016)."

- **Line 193: "within a factor of two" — is that any good? If so, please explain. If not, also please explain. In general give more context to these ranges, otherwise they can be interpreted differently by different people. For example, a factor of two is really bad in my opinion...**

  See answer to previous point: now we have clarified that the comparison to ozone observations is better than previous evaluation (e.g. Jöckel et al., 2016) of the same model.

- **Line 194: delete "somewhat"**

  The word "somewhat" has been removed

- **Add readable legends to Figure 2.**

  We have modified the legend and modified the caption of the figure. Now it reads "RED simulation" and "Observations".

- **Line 206: "hypothesize" — please say more. Can we test this hypothesis? If so, how?**

  The sentence has been removed and text extended in lines 251-255, also following comments from referee #2.

- **Line 212: delete "The vertical ... reproduced (see Fig. 3)." Also "quantitatively" in what sense? Can you qualify that more if you want to keep it?**

  The sentence has been extended explaining what we meant with "qualitatively" : "The vertical profile of the measured aerosol number concentration is qualitatively reproduced (see Fig. ??, with logarithmic scale in the x-axis), with a minimum at $\simeq$ 300 hPa and a maximum at the surface."

- **Lines 223–226: Please either list these volcanoes of interest (obviously super important; you do list some of them later, e.g. Line 243) or don't leave vague language like this around. This could be an opening for you to improve the manuscript anyway**

  The volcanoes of interested are listed in line 257. The sentence in the same line has also been reformulated : "We tested this by injecting high levels of $SO_2$ in the stratosphere in additional simulations, mimicking volcanic eruptions that had enough energy to reach the stratosphere, i.e. Raikoke (June) and Ulawun (June and August) in 2019 (see de Leeuw et al., 2021; Kloss et al., 2021)."

- **Line 232: "We conclude that" — can you give your reasoning to this conclusion? Is it a conclusion anyway or an observation at this stage?**

  We have changed the words "We conclude that [...]" with "We can observe that [...]"

- **Figure 3 like Figure 2 (add legends)**

  Legends have been changed, as Figure 2.

- **Figure 4: please make it bigger and clarify it.**

  The figure is now bigger. The figure is explain in lines 246-251.

- **Line 250: "Results" again, see my above comment about "results"**

  The "Results" word has been removed

- **Line 257: these are not "purely attributable" to differences in your model or are not well captured by it, correct?**

  In this sentence we mentioned that the two model simulations BASE and RED differs ONLY for the emissions, and not for the dynamics. This implies that they experience, for example, the same advection. As results, the differences between these two simulation are ONLY caused by the different emissions, while all other process are identical.

  We rephrase it to make it clearer (line 265): " As no difference in dynamics between RED and BASE simulations are present, any chemical differences between these simulations are purely attributable to the different emissions during the lockdown period, as these are the only changes between these two simulations, and the consequent different chemical regimes."

- **Line 266: "most tracer" should be "most tracers"**

  The word "tracer" has been changed to "tracers" (line 280).

- **Lines 270–271: please elaborate more on this.**

- **Figure 5: add legends and make bigger**

  The legend is present in the uper left panel.

- **Line 284: "monthly mean sulfate (and inorganic aerosols, not show) and black" should be rephrased**

  The sentence has been changed. " Compared to the baseline emissions, the monthly mean sulfate (and inorganic aerosols, not shown) and black carbon concentrations are reduced in all troposphere, with a strong relative reduction at the commercial flight level [. . . ].".

- **Line 286: "lockdown scenario" please unify your naming.**

  We changed the words "lockdown scenario" to " scenario with reduced emissions due to lockdown", analogously to line 153 (where the model simulations are described).

- **Line 299: these are not really close values, are they? They are within the range of error. What's the range of error for the second value btw?**

  This sentence has been removed, following the discussion with referee #2.

- **Line 302: is that significant?**

  Yes, it is significant at 95% confidence level (via t-test).

- **Line 335: "We should note" instead of "We should notice"**

  The text has been changed to "We should note".

- **Line 360: Could you reflect on this range a little more? Seems insignificant and uncertain to a casual reader.**

  We have extended the text with the following lines: " The differences between these studies, can partly be attributed to the applied methodologies and general difficulties in discriminating anthropogenic effects from interannual variability. Hence, a study which considers contrail and aerosol effects simultaneously, and covers a longer time period, is recommended to better attribute the causes of the observed changes ."

- **Code availability: Why list all these details about doing MOU and all that, can you just give the git repository link and tag/commit?**

  Unfortunately, due to the contained legacy models, we are not allowed to host parts of the code open source or publicly accessible. Our GitLab repository is not open and a URL would be therefore meaningless. This section is standard and used in the sister journals GMD(D) as mentioned in the license section of the MESSy system (https://www.messy-interface.org/).

- **Data availability: Please make your data available, and refrain from "contact the author" stuff. It doesn't seem open...**

  The observational data and the model results are available on the HALO (High Altitude Long RAnge research aircraft) database (https://halo-db.pa.op.dlr.de), upon sign of data protocol. This was included in the revised manuscript.

[revised manuscript text omitted]